# Imputed genomes and haplotype-based analyses of the Picts of early medieval Scotland reveal fine-scale relatedness between Iron Age, early medieval and the modern people of the UK

**Adeline Morez**[1]*, **Kate Britton**[2,3], **Gordon Noble**[2], **Torsten Günther**[4], **Anders Götherström**[5], **Ricardo Rodríguez-Varela**[5], **Natalija Kashuba**[6], **Rui Martiniano**[1], **Sahra Talamo**[3,7], **Nicholas J. Evans**[2], **Joel D. Irish**[1], **Christina Donald**[8], **Linus Girdland-Flink**[1,2]*

1 School of Biological and Environmental Sciences, Liverpool John Moores University, Liverpool, United Kingdom, 2 Department of Archaeology, School of Geosciences, University of Aberdeen, Aberdeen, United Kingdom, 3 Department of Human Evolution, Max Planck Institute for Evolutionary Anthropology, Leipzig, Germany, 4 Department of Organismal Biology, Uppsala University, Uppsala, Sweden, 5 Department of Archaeology and Classical Studies, Stockholm University, Stockholm, Sweden, 6 Department of Archaeology and Ancient History, Uppsala University, Uppsala, Sweden, 7 Department of Chemistry, University of Bologna, Bologna, Italy, 8 The McManus: Dundee's Art Gallery and Museum, Dundee, United Kingdom

* adelinemorez@gmail.com (AM); linus.girdlandflink@abdn.ac.uk (LGF)

**Data Availability Statement:** All raw and mapped sequence data generated for this project are

## Abstract

There are longstanding questions about the origins and ancestry of the Picts of early medieval Scotland (*ca.* 300–900 CE), prompted in part by exotic medieval origin myths, their enigmatic symbols and inscriptions, and the meagre textual evidence. The Picts, first mentioned in the late 3rd century CE resisted the Romans and went on to form a powerful kingdom that ruled over a large territory in northern Britain. In the 9th and 10th centuries Gaelic language, culture and identity became dominant, transforming the Pictish realm into Alba, the precursor to the medieval kingdom of Scotland. To date, no comprehensive analysis of Pictish genomes has been published, and questions about their biological relationships to other cultural groups living in Britain remain unanswered. Here we present two high-quality Pictish genomes (2.4 and 16.5X coverage) from central and northern Scotland dated from the 5th-7th century which we impute and co-analyse with >8,300 previously published ancient and modern genomes. Using allele frequency and haplotype-based approaches, we can firmly place the genomes within the Iron Age gene pool in Britain and demonstrate regional biological affinity. We also demonstrate the presence of population structure within Pictish groups, with Orcadian Picts being genetically distinct from their mainland contemporaries. When investigating Identity-By-Descent (IBD) with present-day genomes, we observe broad affinities between the mainland Pictish genomes and the present-day people living in western Scotland, Wales, Northern Ireland and Northumbria, but less with the rest of England, the Orkney islands and eastern Scotland—where the political centres of Pictland were located. The pre-Viking Age Orcadian Picts evidence a high degree of IBD sharing across modern

available from the European Nucleotide Archive under the study accession number PRJEB58104. All imputed data generated for this project are available from the European Variation Archive under the study accession number PRJEB60907.

**Funding:** AM was supported by ECR strategic support of early career researchers in the faculty of science at LJMU, awarded to LGF. LGF was supported by the School of Geoscience, University of Aberdeen. KB was supported by the Leverhulme Trust (PLP-2019-284) during production of this manuscript. The funders had no role in study design, data collection and analysis, decision to publish, or preparation of the manuscript.

**Competing interests:** The authors have declared that no competing interests exist.

Scotland, Wales, Northern Ireland, and the Orkney islands, demonstrating substantial genetic continuity in Orkney for the last ~2,000 years. Analysis of mitochondrial DNA diversity at the Pictish cemetery of Lundin Links (n = 7) reveals absence of direct common female ancestors, with implications for broader social organisation. Overall, our study provides novel insights into the genetic affinities and population structure of the Picts and direct relationships between ancient and present-day groups of the UK.

## Author summary

We report two high-quality autosomal and eight mitochondrial genomes sequenced from individuals associated with the Pictish period of early medieval Scotland (*ca*. 300–900 CE). We demonstrate genetic affinities between the Pictish genomes and Iron Age people who lived in Britain, which supports current archaeological theories of a local origin. The autosomal genomes also allowed us to detect haplotype sharing between the Pictish genomes and present-day Europeans. Our results demonstrate a proportionally higher degree of haplotype sharing, and thus genetic affinity, between the Pictish genomes and individuals from western Scotland, Wales, Northern Ireland and Northumbria. We also detected genetic structure in Scotland during the Iron Age, likely driven by the combination of genetic drift and small population size, which we also detect in present-day Orcadians. Lastly, the seven mitochondrial DNA from the Lundin Links cemetery showed that these individuals had no direct maternal ancestors which could suggest exchanges of people, or at least females, between groups during the Pictish period, challenging older ideas that the Picts were a matrilineal society. Overall, our results show that high-quality ancient genomes combined with haplotype imputation are highly informative for obtaining novel insights to population structure and migration over the past 2,000 years.

## Introduction

The genetic origins of the present-day populations of the UK have been extensively studied and can broadly be modelled as a mixture of three deep genetic ancestries, mirroring western European ancestry: western European Mesolithic hunter-gatherer ancestry, Early European farmer ancestry derived from Anatolian Neolithic farmers, and Late Neolithic steppe-related ancestry [1–10]. Our understanding of more recent demographic changes in the British and Irish Isles–referring to Britain, Ireland and associated smaller islands–has also been expanded via large-scale sequencing of ancient genomes, revealing extensive gene flow from mainland Europe into southern Britain during the Middle Bronze Age, which contributed to genetic differentiation between Iron Age groups from southern and northern Britain [9, 10]. Present-day genetic diversity in Wales, Cornwall, Devon and western Ireland indicates a long-standing genetic structure, possibly already present during the Iron Age [11], but the lack of ancient samples especially from Scotland limits our ability to directly test this hypothesis, and 'pockets' of older ancestries could have survived regionally in isolated populations for extended periods [12,13].

The British and Irish Isles witnessed a complex cultural turnover from the Iron Age to the early medieval period. The Romans occupied part of Britain to southern Scotland from 43 to *ca*. 410 CE; however, the single aDNA study of Roman Britain suggests this occupation resulted in little detectable gene flow from mainland Europe [14]. Multiple episodes of long-

distance migration across western and central Eurasia [9,15,16] intensified during the Late Antique period (*ca.* 300–800 CE), before and following the collapse of the Western Roman Empire. In Britain, Angles, Saxons and other Germanic-speaking peoples, likely originating in Scandinavia, the Low Countries and parts of Germany, settled predominantly in south-eastern and central Britain with genetic evidence of extensive admixture with local populations carrying genetic ancestry from the Iron Age [14,17,18]. During the so-called 'Viking Age' (starting about 800 CE), Scandinavians settled in the 'Danelaw' in northern and eastern England, as well as in the coastal areas of Ireland and northern and western Britain [19], which led to admixture with the inhabitants of Ireland and western and northern Britain over nearly four centuries [9]. In addition, local, culturally distinct groups lived in Britain around the end of the Roman period, before major Anglo-Saxon settlement: the Britons (speaking the ancestral language of Welsh, as well as Latin) inhabited the island south of the Firth of Forth, the Gaels (Dál Riata) occupied Argyll and the southern Hebrides in Scotland, and the Picts lived in the rest of Britain north of the Forth [20,21]. The genetic diversity between and within these groups is poorly understood. In particular, the lack of genomes from Scotland has limited our ability to understand how the genetic structure changed between the Iron Age and the early medieval period.

Among the peoples present during the first millennium CE in Britain, the Picts (*ca.* 300–900 CE) are one of the most enigmatic. Their unique cultural features (e.g. Pictish symbols) and the scarcity of direct writing resulted in many diverse hypotheses about their origin, lifestyle and culture, the so-called 'Pictish problem' [22]. Other than a list of kings and difficult-to-decipher ogham and alphabetic inscriptions, the only written evidence comes to us from their neighbours–the Romans and later the Gaels, Britons and Anglo-Saxons. This deficiency has been compounded by a sparse archaeological record with few settlements and fewer cemeteries from this period [23].

In the modern era perceptions of Pictish origins have varied, often according to cultural and political biases, with the Picts and their languages regarded as Germanic, Gaelic, Brittonic, Basque, and Illyrian, among other theories. In the 1950s Jackson influentially argued that the Picts spoke a non-Indo-European language and a Celtic language akin to ancient Gaulish [24,25]. The current consensus is that they spoke a Celtic language closest to that of neighbouring Britons from which Cornish, Welsh, and Breton derive [26–28]. However, some still argue from undeciphered inscriptions and other words that some Picts spoke an otherwise unknown language, presumably derived from a pre-Celtic population [29, 30]. This theory is underpinned by recent findings arguing that Late Bronze Age migrations in southern Britain may have introduced Celtic or Celtic-related languages in this area; however, these migrations were seemingly not as influential in northern Britain [10]. Thus, the question remains of whether the Picts were somehow fundamentally different from their neighbours.

In the medieval period, the Picts were considered immigrants from Thrace (north of the Aegean Sea), Scythia (eastern Europe), or isles north of Britain [31]. However, Irish accounts and the Northumbrian scholar Bede added that, before settling in Britain, the Picts first gained wives in Ireland, on the condition that Pictish succession passed through the female line. This is the origin of the theory that the Picts practised a form of matriliny, with succession and perhaps inheritance going to the sister's son rather than directly through the male line. However, our earliest source for this practice, Bede's 'Ecclesiastical History of the English People' (finished in 731 CE), stated that it was limited to occasions when the succession was in dispute. It is now argued that the origin-legend was intended to reinforce Pictish identity and legitimise particular kings whose claims to the throne were through their mothers [32, 33]. Nevertheless, matriliny remains one potential explanation for the absence of father-to-son succession in at least one Pictish royal dynasty before the mid-8[th] century [34].

Here, we aim to provide new insights into the genetic diversity of the Picts through analysis of two ancient whole genomes (2.4 and 16.5X coverage) sequenced from individuals excavated from two Pictish-era cemeteries: Lundin Links (Fife, Southern Pictland) and Balintore (Easter Ross, Northern Pictland). We imputed diploid genotypes alongside published medium-to-high coverage ancient genomes, including individuals from England dating from the Iron Age, Roman and early medieval periods [14,17], and co-analysed these with previously imputed ancient genomes from the Orkney islands dating from the Late Iron Age (Pictish period) and Viking Age [9]. Using allele frequency and haplotype-based methods, we aim to determine the genetic relationships between the Picts and neighbouring modern-day and ancient populations. In addition, using the mitogenomes of seven individuals from Lundin Links, we explore how differences in female mobility due to possible post-marital residence customs, i.e., matrilocality (female endogamy), may have shaped the genetic diversity at this possible high-status cemetery and discuss its implication for our understanding of Pictish elite descent systems.

## Results and discussion

### DNA extraction, sequencing, and radiocarbon dating

We retrieved DNA from eight individuals, one from Balintore (Easter Ross) and seven from Lundin Links (Fife), representing the northern and southern parts of Pictland (Fig 1 and Tables 1 and S1 and section S1.1 in S1 Text). Two individuals, BAL003 and LUN004 were treated with the USER enzyme to remove post-mortem deamination [35] and were shotgun sequenced to medium and high coverage (2.4 and 16.5X) (Tables 1 and S1). Seven individuals from Lundin Links were shotgun sequenced to a sufficient mitochondrial DNA (mtDNA) coverage for subsequent analyses (3.47–195.05X, Table 1). LUN001 and LUN003 are excluded from the population genetics analysis involving autosomal DNA as we found evidence for library index misassignment in the autosomal data, but not the mitochondrial data (section S1.2 in S1 Text). Ten samples from Lundin Links (including LUN001, LUN002, LUN003 and LUN009) and three from Balintore were radiocarbon dated to the 5th-7th century CE (Tables 1 and S4 and S2 Fig and section S1.1 in S1 Text) [36]. The individuals will be referred to as Pictish throughout the text as they lived during the period that Pictish identity existed and are from areas likely to have been within Pictish territories *ca.* 700 CE.

All samples evidenced deamination patterns at the fragment termini characteristic of ancient DNA (aDNA), except BAL003 and LUN004 which underwent damage repair enzymatic treatment, for which the respective unrepaired screening libraries BAL003-b1e1l1 and LUN004-b1e1l1 did show deamination (S3 Fig). Contamination estimates based on mtDNA, X and Y-chromosome reads are low (Tables 1 and S1 and section S1.3 in S1 Text). The genetic sex determination (S4 Fig) agrees with the morphological sex determination (Tables 1 and S1).

### Demographic history

**Analysis of uniparental genetic markers.** The mitochondrial haplogroups observed in the samples are common in present-day north-western Europeans, with the sub-clade J1c3 being identified in three individuals out of eight (Tables 1 and S5). In terms of paternal Y-chromosomal lineages, we assigned LUN004 to R1b-DF49 (Tables 1 and S6), which is predominantly distributed in the UK and Ireland and a sub-clade to R1b-P312/S116 haplogroup introduced to Britain by Bell Beaker peoples during the Chalcolithic, alongside steppe-related ancestry [7]. During the Chalcolithic, R1b-derived haplogroups largely replaced the predominant I2a Y-chromosome lineage in the British Neolithic [7,8,40] except in Orkney where I2a persisted into the Bronze Age [13]. R1b sub-clades are extremely common across Britain and western Europe from the Iron Age onwards [13,14,17,41].

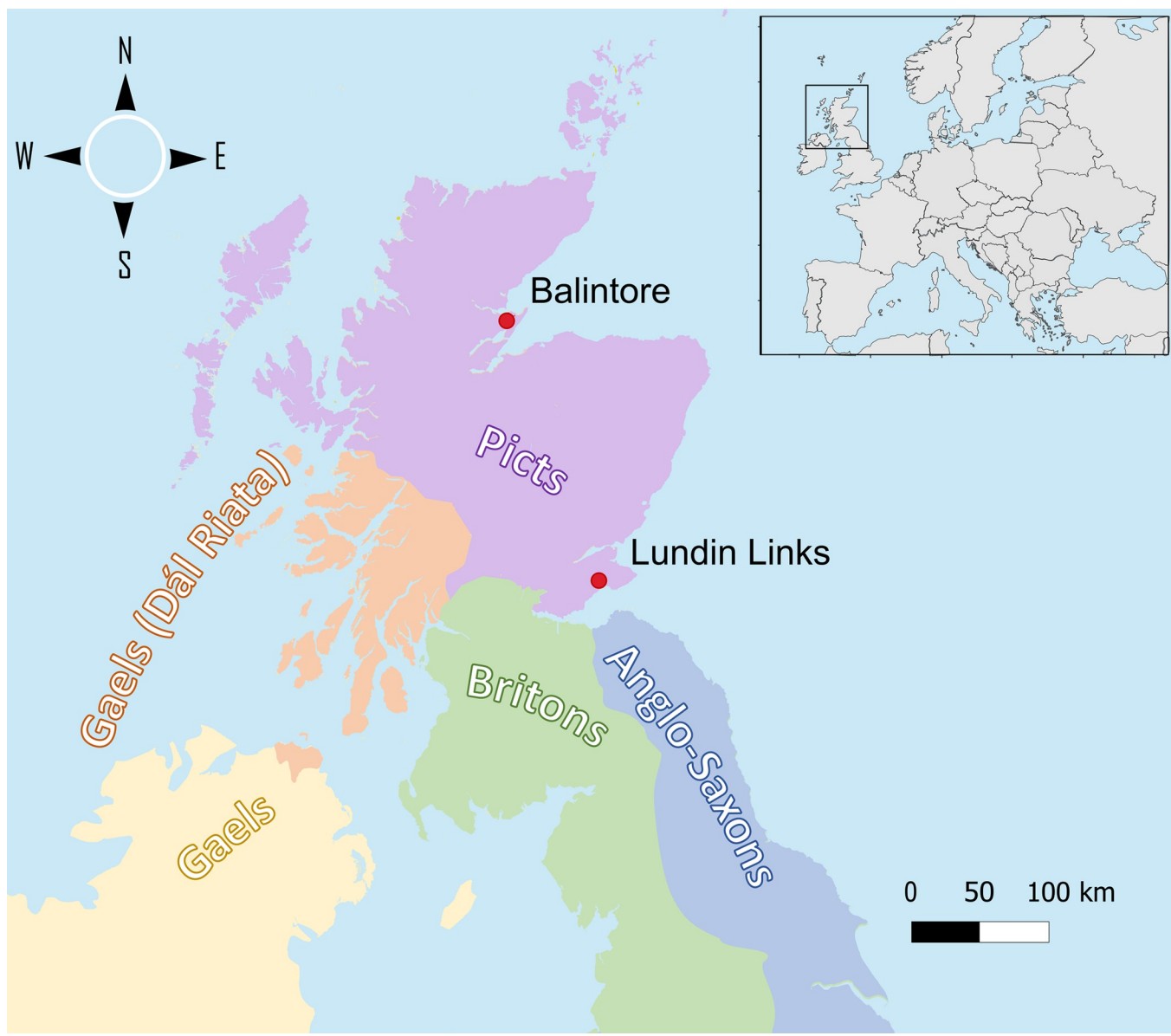

**Fig 1. Sampling location and the regions under ancient Brittonic, Irish and Anglo-Saxon control around the 7th century [37–39].**

**Allele frequency-based genomic affinities in ancient Britain.** To investigate population affinities of the individuals from Pictland, we performed Principal Component Analysis (PCA) and ADMIXTURE analyses on a dataset comprising present-day Europeans, the newly imputed genomes and the imputed ancient genomes from Margaryan et al. [9] (S7 Table). The PCA shows that the ancient individuals from Britain broadly fit with present-day individuals whose four grandparents lived in Britain (Fig 2A). However, we notice some variability among these individuals as BAL003 and LUN004 fall within the modern Welsh cluster, but with BAL003 being notably closer to the present-day Scottish, Orcadians, English and Northern Irish clusters, suggesting some degree of genetic differentiation amongst individuals from Pictland. The Iron Age and Roman period individuals from England are spread across the modern English, Northern Irish, Scottish and Welsh clusters. Four ancient Orcadians from the Iron

**Table 1. Summary information from the eight samples investigated in this study.**

| Sample | Site | Radiocarbon calibrated date (95.4% confidence, AD) | Genome coverage (X) | mtDNA Coverage (X) | Sex | mtDNA / Y haplogroup | Contamination Estimate (%) | | |
|--------|------|---------|---------|---------|-----|---------|---------|---------|---------|
| | | | | | | | Based on mtDNA | Based on X | Based on Y |
| BAL003 | Balintore | 419–538 | 16.54 | 294.97 | XX | H2a1e | 1 ± 1 | - | 0.30 |
| LUN001 | Lundin Links | 416–545 | 0.18 | 195.05 | XX | T2a1a | 2 ± 1 | - | 3.78 |
| LUN002 | Lundin Links | 563–653 | $2.24 \times 10^{-3}$ | 3.47 | - | H1c20 | 1 ± 1 | - | - |
| LUN003 | Lundin Links | 384–562 | 0.32 | 154.31 | XX | T2b11 | 2 ± 1 | - | 1.70 |
| LUN004 | Lundin Links | 406–542 | 2.43 | 121.32 | XY | J1c3g / R1b- DF49 | 1 ± 1 | 0.80 ± 0.10 | - |
| LUN005 | Lundin Links | | $7.02 \times 10^{-3}$ | 3.99 | XY | K1c2 | 1 ± 1 | - | - |
| LUN006 | Lundin Links | | $1.48 \times 10^{-3}$ | 8.13 | - | J1c3b1 | 2 ± 1 | - | - |
| LUN009 | Lundin Links | 430–590 | $6.11 \times 10^{-3}$ | 4.26 | XX | J1c3b | 1 ± 1 | - | 0.59 |

and Viking Ages fit with present-day Welsh, Northern Irish and Scottish populations. However, two Viking Age Orcadians (VK204 and VK205) are intermediates between the British and Scandinavian clusters, consistent with previous results finding evidence of admixture in these individuals between British-like and Scandinavian-like ancestries [9]. The early medieval individuals from England are intermediate between modern English people and Scandinavians, which is consistent with various degrees of admixture between Iron Age groups from England and immigrants from northern/central Europe [14,17,18]. These results agree with the pseudo-haploid-based analyses of the BAL003 and LUN004 genomes, showing a broad affinity to modern western Europeans (section S1.3 in S1 Text and S10, S12–S15 and S18 Figs), but with a much-improved resolution.

## Haplotype-inferred genetic structure in early medieval Britain

Haplotype-based methods have been shown to outperform conventional unlinked SNP approaches in the detection of population substructure [42–44]. To make use of the additional power provided by haplotype information, we conducted a FineSTRUCTURE clustering analysis and Identity-By-Descent (IBD) analysis on the imputed diploid dataset. Our analysis shows that the genomes from Lundin Links and Balintore form a genetic cluster together with genomes from the Iron Age and Roman period from England (except 6DT3 –who instead shows strong affinity to western/central Europe or Scandinavia based on the IBD analysis, Fig 3 and section S1.6 in S1 Text–and I0160), and from the Late Iron Age to Viking Age from Orkney (except VK204 and VK205 who carried substantial Scandinavian-like ancestry; 'Pop12', Figs 3 and S27). Included in this cluster are also Viking Age individuals from Britain, Iceland and Scandinavia; the latter likely corresponds to individuals buried in Scandinavia but whose parents were from a British-like gene pool, consistent with results in Margaryan et al. [9]. Based on *outgroup-f3*, the individuals from Orkney, Scotland, and England, dated from the Iron Age to medieval period are symmetrically related to each other (S5 Fig). However, we also note that BAL003, but not LUN004, show multiple instances of IBD sharing >4 cM with early medieval individuals from England (S21 Fig), which is also reflected in their relative position in the PCA (Fig 2A), implying substantial shared ancestry and possibly recent gene-flow

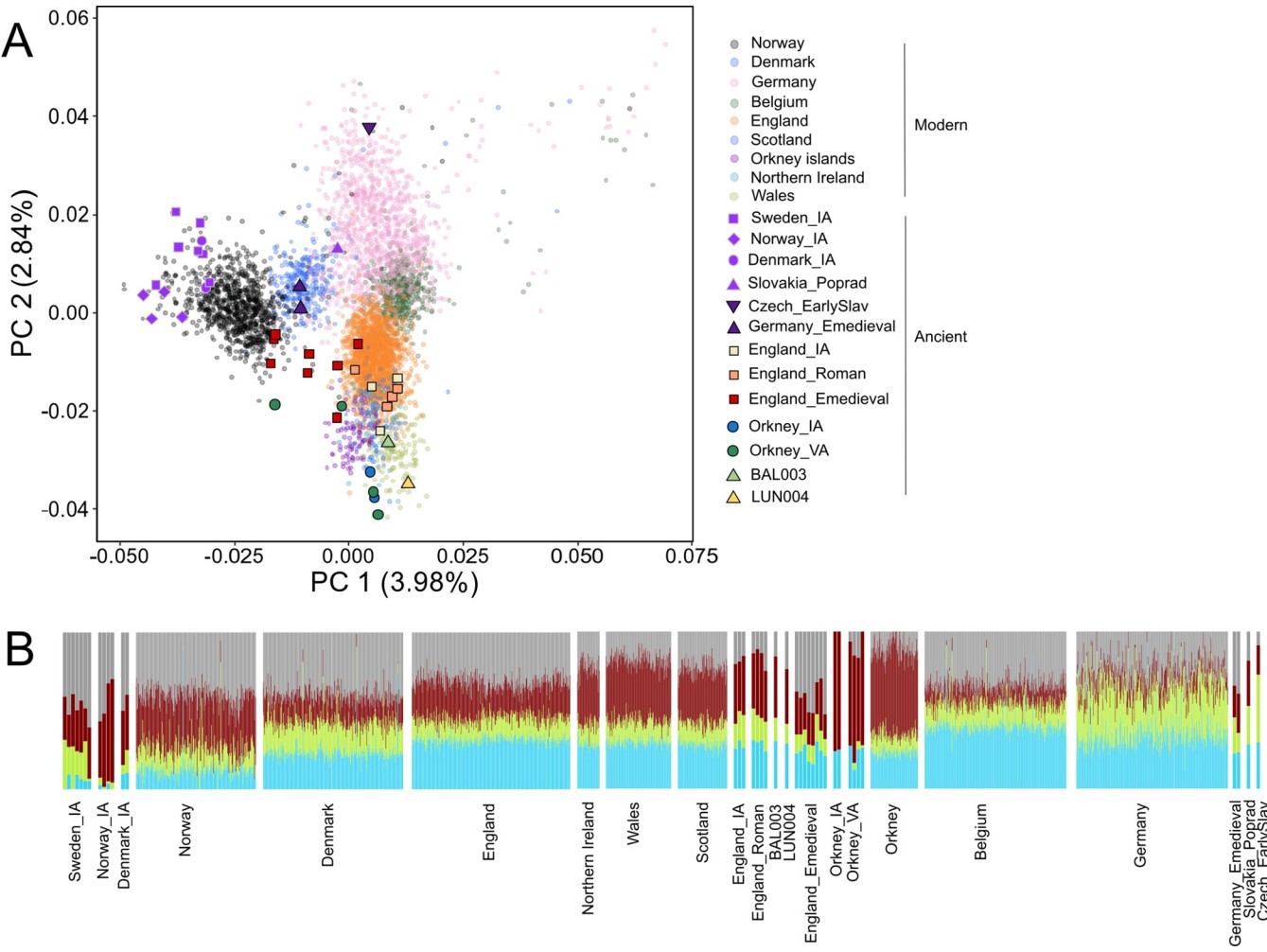

**Fig 2. Genetic diversity of Iron Age, early medieval and present-day individuals from northern and central Europe. A)** A Principal Component Analysis of 4,914 individuals and 87,518 SNPs. **B)** An ADMIXTURE ancestry component (K = 4) of these same genomes (see S7 Fig for the complete analysis from K = 2 to K = 10).

from a source genetically similar to those samples. We therefore suggest that individuals from Pictland should not be considered a homogenous genetic group, but instead a complex mixture of contemporary genetic ancestries.

The unlinked approach implemented in the ADMIXTURE analysis also reveals a minor but detectable genetic structure consistent with results from the PCA (Fig 2A) but not evident in the FineSTRUCTURE analysis (Figs 2B and S7). While the proportion of ancestry components are similar across BAL003, LUN004, Iron Age and the Roman period in England, the Late Iron Age and unadmixed Viking Age Orcadians are differentiated from this group (Figs 2B and S7). They show an absence of the grey and green ancestry components, likely first introduced by Scandinavian migrants as they are first observed in VK204 and VK205 and then in modern Orcadians. These components are also carried at a high proportion in modern Norwegians and Danes. Iron and Viking Age individuals from Orkney with an absence of Scandinavian-like admixture share more genetic drift with modern Orcadians [0.08 < *f3(Orkney Iron or Viking Age (unadmixed), modern Orcadians; Yoruba)* < 0.13] than do BAL003 and LUN004 [*f3(Picts, modern Orcadians; Yoruba)* = 0.03]; we calculated this on ancient pseudo-haploid

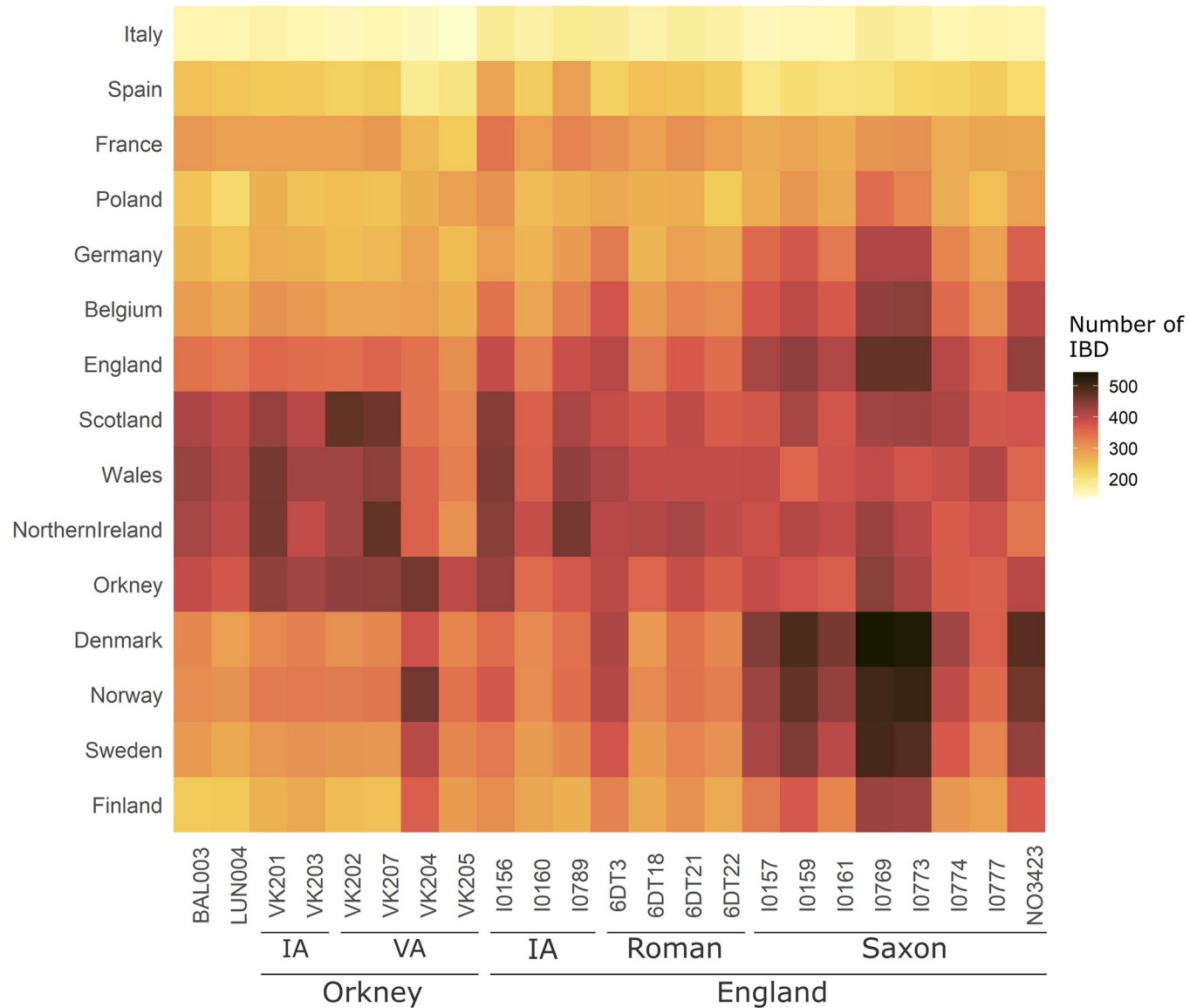

**Fig 3. Shared Identity-By-Descent (IBD) segments >1 cM between the ancient genomes from Britain and present-day European populations.** IA, Iron Age. VA, Viking Age.

genomes to avoid allele frequency bias originating from the imputation. The results confirm the relatively high proportion of shared genetic drift in Orkney for the last 2,000 years, which supports our interpretation of the ADMIXTURE plot. Nevertheless, a high relative count of IBD sharing (>1 cM, >4 cM and >6 cM) between LUN004 and Late Iron Age or Viking Age Orcadians (S21 Fig) demonstrates that gene flow between Orkney and mainland Scotland–or a source outside Scotland that also contributed ancestry to LUN004–likely occurred.

The Iron Age and unadmixed Viking Age Orcadians also show the highest degree of a red ancestry component (Figs 2B and S7), which is inconsistent with having originated from direct gene flow from any population included in this study and instead likely reflects retention of a less diverse pre-Iron Age ancestry in Orkney and/or strong genetic drift (such as a bottleneck or founder effect) (see [45] for insight on how a bottleneck or admixture can result in a similar

ADMIXTURE profile). In fact, recent research show that Bronze Age populations in Orkney were differentiated from their counterparts on mainland Britain because of strong genetic drift as a result of a small ancestral population size, which was deduced from the observation of a relatively high number of short runs of homozygosity [13]. The Bronze Age Orkney population also retained the Y-chromosomal haplogroup I2, which is associated with Neolithic ancestry, while the R1b haplogroup associated with Bell Beaker expansion largely replaced the I2 haplogroups in the rest of Britain [13]. Moreover, although modern Orcadians are differentiated from the rest of the British Isles due to extensive admixture with Scandinavians, recent genomic research shows that genetic drift also played an important role [11,46]. This is consistent with our results that show a high proportion of shared IBD segments among modern Orcadians (>1 to >6 cM, S19 Fig), meaning they share a high proportion of recent common genetic ancestors relative to most modern European populations, typical of small or genetically isolated populations. Three Orcadians dated from the Late Iron Age and Viking Age also displayed the highest number of small HBD <1.5 cM (S26 Fig), typical of individuals descending from a small population. One ancient individual from Orkney (VK201) evidenced a long Homozygosity-By-Descent (HBD) segment (>9.5 cM), the longest observed amongst all ancient individuals and indicative of small population size or consanguinity (S26 Fig). Overall, these data indicate a long-term small population size, which likely contributed to the extensive genetic drift observed in modern Orcadians. The genetic differentiation between populations living in Orkney and Scotland during the Late Iron Age and early medieval period could thus be partially explained by different degrees of genetic drift.

## Analysis of genetic continuity across Britain

The Pictish data allow us to obtain a transect of Iron Age/early medieval genomes across Britain and directly look at the pattern of haplotype sharing between them and present-day genomes. The Iron Age and Roman period (except 6DT3) individuals from England and Scotland share more IBD segments >1 cM (both in terms of number and length) with present-day individuals from Scotland (including Orkney), Northern Ireland and Wales than with any other European populations included in our analyses (Figs 3 and S20), consistent with the structure observed in the PCA analysis (Fig 2A). We also show that all early medieval individuals (excluding I0777) share more IBD with modern Danish than with any other present-day population (Fig 3), suggesting genetic continuity between modern-day Danish and the ancestors of these individuals (section S1.6 in S1 Text).

The analysis also revealed high IBD sharing between early medieval individuals from England and present-day people across Britain following a southeast/northwest cline (Figs 4 and S22). This pattern suggests that northern continental European ancestry associated with Anglo-Saxon migrations expanded out of south-eastern England followed by admixture with local populations, a scenario consistent with previous research [11,14,17,18,46,47]. BAL003 and LUN004 share a high proportion of IBD segments with present-day people from western Scotland, Wales and Northern Ireland, similar to the individuals from Late Iron Age Orkney and England (Figs 4 and S22). However, unlike these individuals, LUN004, and to a lesser extent BAL003, shares fewer IBD segments with the present-day eastern Scottish population sample (Figs 4 and S22). Byrne et al. [47] and Gilbert et al. [46] previously suggested that the genetic structure between western and eastern Scotland could result from the divide between the kingdoms of the Gaelic-speaking Dál Riata in the west and Picts in the east, which is seemingly in contradiction with the results presented here. Instead, the present-day genetic structure in Scotland likely results from more complex demographic processes that cannot be reduced to a single model.

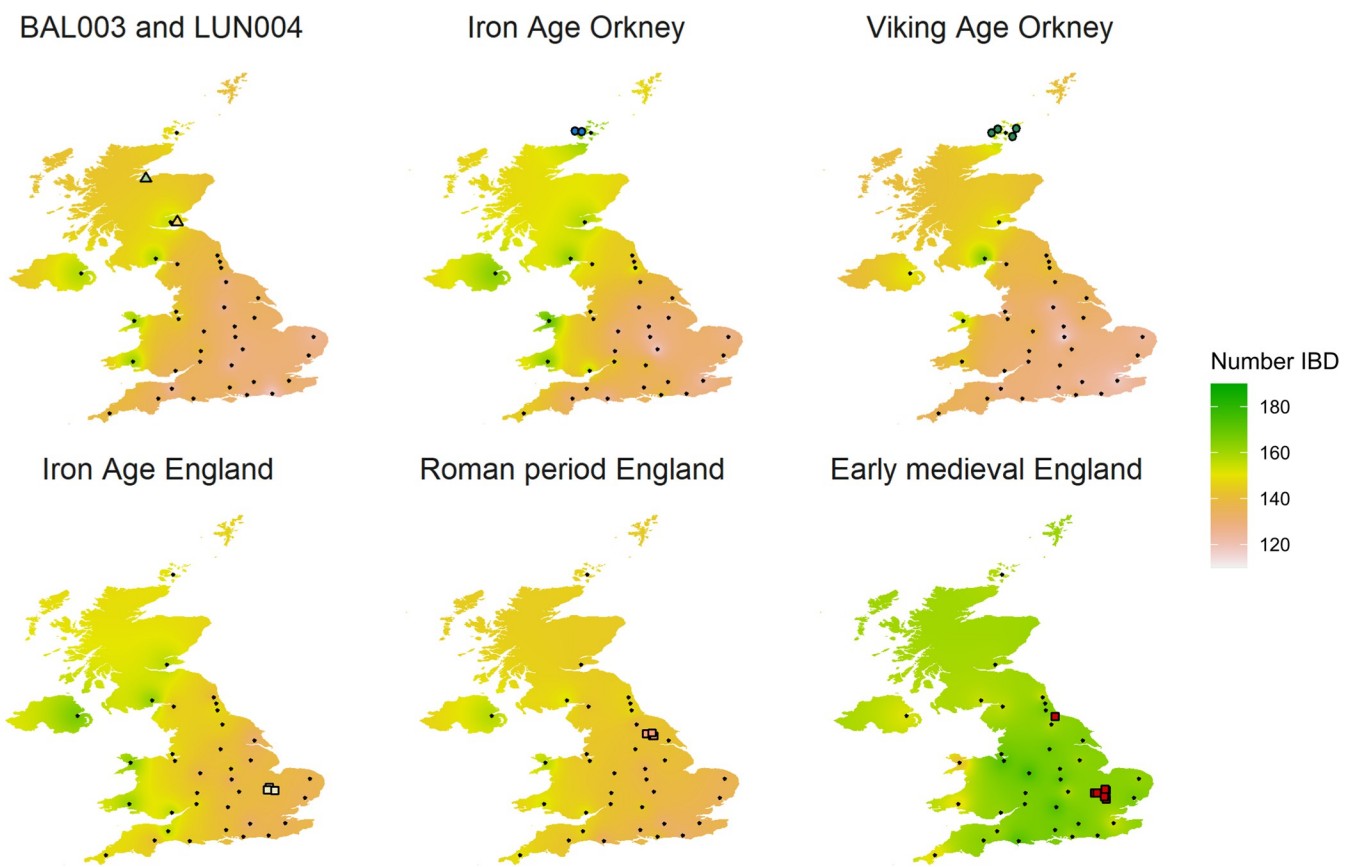

**Fig 4. Average IBD sharing >1 cM between present-day and ancient groups from the UK.** IBD sharing between each of the ancient genomes and modern samples is illustrated in S22 Fig. Ancient individuals are indicated with coloured symbols. The black dots represent the geographic location of present-day people from 35 regions of the UK [11, 53], by the county town.

We propose two non-exclusive processes that might explain the observed pattern of IBD sharing between the Iron Age and early medieval populations and the present-day Scottish population. The first is substantial admixture from immigrants that brought Iron Age Orcadian-, and England-like ancestries (likely independently), which partially replaced the eastern Scottish early medieval gene pool. Indeed, in the following centuries (1,100–1,300 CE), eastern Scotland received substantial immigration, such as settlers from Britain south of the Forth, France, and the Low Countries [48–50]. Under this scenario, BAL003 and LUN004 are good representatives of the broader ancestry present in Scotland during the Pictish period. Alternatively, the ancestors of BAL003 and LUN004 share more IBD segments with present-day people from western Scotland, Wales, and Northern Ireland because they (or their direct ancestors) migrated from these regions but did not contribute substantially to later generations via admixture with local groups in eastern Scotland. This scenario is consistent with an emerging picture of west-east lifetime mobility of both males and females in the early medieval period in Scotland [51,52]. Under such a model, it may be feasible that there are indeed still undiscovered 'pockets' of eastern Pictish-period ancestry, that was differentiated from ancestry carried by BAL003 and LUN004 and which contributed significantly to present-day populations from eastern Scotland. Oxygen and strontium isotope analysis of teeth from these individuals holds promise to characterise this further. Importantly, we also emphasise that

stochasticity likely affected the pattern of IBD sharing in such a small sample size. Indeed, high variability in IBD sharing is observed amongst individuals from the early medieval and Iron Age groups, and to some extent between BAL003 and LUN004 (S22Fig).

Our results also show substantial IBD sharing between Iron Age, Viking Age and present-day Orcadians, supporting our observations using allele-frequency based methods of strong genetic continuity in this region over time (Figs 2, 4, and S22). Therefore, the marked genetic differentiation between Orkney and mainland Britain is not only a result of Scandinavian admixture, as previously hypothesised [11,46,54–57] but also pronounced genetic continuity that persisted for at least 2,000 years. The relatively low IBD sharing between BAL003 and LUN004 and modern-day Orcadians (Fig 4) suggests the emergence of Pictish culture in Orkney [22,23,37] was not associated with population replacement but largely due to cultural diffusion and connections.

IBD segments in Iron Age individuals from south-eastern England are widespread throughout western and northern Britain compared to the more recent Romano-British individuals from northern England; the latter, however, do not share substantial IBD with any present-day people of the British Isles (Figs 4 and S22). The only exception is 6DT3 who was from the same genetic population as two early medieval individuals (I0159 and I0773) with Scandinavian-, and northern European-like ancestry ('pop12', S22 Fig and section S1.6 in S1 Text). 6DT3 also share relatively more IBD segments >1 cM with the present-day population from Scandinavia, Belgium and the UK (Fig 3), suggesting that Scandinavian-like ancestry could have spread to the British Isles before the early medieval period.

These results need to be interpreted with caution because of existing flaws in the IBD segment detection. First, it has previously been demonstrated that the false positive rate (genomic regions falsely attributed to IBD segments) is high, especially for small IBD segments [43]. They are usually created from the concatenation of shorter IBD segments into a false longer one. The false positive rate is around 10–40% for ≤2 cM IBD segments, but decreases with the length of the segment and is close to 0% for IBD segments ≥5 cM. Several factors can influence this rate, the main ones being data quality and population history. The detection of false positives varies with the density and informativeness of genetic markers. For example, segments of the genome containing many rare alleles are easier to identify as IBD. Regions with low SNP density will be associated with a higher false positive rate. Ralph and Coop [43] also detected significant variation in the false positive rate between populations for IBD segments <2 cM, with populations from Spain, Portugal and Italy showing significantly higher false positive rates than other European populations. A likely explanation given by the authors is that IBD detection algorithms are based on haplotype frequency, so the false positive rate should increase in populations that are more differentiated from the rest of the sample; in this case, they are the populations sharing the least common genetic ancestors both within their respective populations and with other European populations. A second flaw is the low power to detect true small IBD <2 cM, as demonstrated in [43] using fastIBD [58]. They estimate that the power of detecting <2 cM IBD segments is <50%. However, in the present study, we used RefinedIBD [59], which was shown to be more powerful in detecting true 1–2 cM IBD segments, reaching 40% power for 1 cM IBD segments (with segment LOD >3), against only 20% using fastIBD. One possible reason that can break down haplotypes is genotyping error [43, 60]; however, we merged segments having at most one discordant homozygote <0.6 cM apart (see section S1.6 in S1 Text) to overcome this issue. Other effects could influence IBD detection, such as a higher chance of difference in recombination rates among ancient samples and the HapMap dataset or presence of structural variants [61].

## Social organisation

Seven mtDNA genomes were retrieved at Lundin Links, which allows us to address questions about the Pictish social organisation reflected in the individuals interred at the site. The use of the cemetery was relatively short, likely around 130 years (S2 Fig), and the individuals excavated were adults (S1 Table) [36]. The diversity of mtDNA lineages was high, and none of the individuals shared an immediate maternal ancestor (S5 Table). It is worth noting that individuals retrieved from the horned cairns complex (S1 Fig) show evidence of familial links based on the presence of highly heritable skeletal traits (congenitally missing teeth, torus mandibularis, and septal apertures on the humeri) [35]. However, the two individuals yielding mtDNA (LUN001 and LUN009) are not maternally related (S5 Table). In a matrilocal system, which is typical of matrilineal descent, low female post-marital migration and high male migration decrease female mtDNA diversity [62–65]. This result suggests the individuals buried at Lundin Links were unlikely to have been practicing matrilocality; this assumes that this cemetery is a snapshot of the community near Lundin Links, and individuals did not return to their natal place for burial after having lived in their marital community. Ongoing isotope analyses focused on the movement histories of the Lundin Links individuals using strontium, oxygen and other isotope approaches may further characterise sex-specific mobility. Additional Y-chromosome analyses will also help confirm whether patrilocality or neolocality was more common in Pictish society [62].

Seventy per-cent of matrilocal societies are associated with a matrilineal system [66]. Insofar as funerary treatment can tell us about the social organisation in life, it is unlikely that the community at Lundin Links followed a matrilineal inheritance system. This interpretation challenges older arguments for matrilineal succession among Pictish rulers [34]. However, while some individuals buried at Lundin Links may have been of elevated social status, the relationship between people buried in monuments such as these and the Pictish uppermost elite is uncertain. The cemetery evidences a wide diversity of cultural practices [36], mirrored in the high mitochondrial diversity, suggesting relatively high levels of mobility within the Pictish social structure at this level of society. The burials are organised in complex and stand-alone graves, made of round and square cairns and long cists (S1 Fig). This complexity suggests that, as social practices influence the genetic structure of populations, the social status of archaeological sites can, in turn, bias our understanding of population structure; in this case, the samples may only be representative of a small proportion of the overall Pictish population. Non-harmonious kinship systems (i.e., patrilocal and patrilineal or matrilocal and matrilineal societies) may also impact the genome in different ways. The lack of broad sample size and useful markers (Y-chromosome) to enhance kinship- and mtDNA-based findings remains an obstacle to illuminate further Pictish descent patterns.

## Conclusions

Our study provides novel insight into genetic affinity between ancient and modern populations of the British Isles, a rare opportunity to directly observe micro-scale evolution. High-quality genomes of two individuals buried in Scotland from the Pictish period, one from Balintore (BAL003) and one from Lundin Links (LUN004), reveal a close genetic affinity to Iron Age populations from Britain but with evidence of some genetic differentiation between samples. Overall, our data supports the current archaeological consensus arguing for regional continuity between the Late Iron Age and early medieval periods, but likely with complex patterns of migration, lifetime mobility and admixture. We also show that BAL003 and LUN004 were genetically differentiated from the pre-Viking Age Picts from Orkney, which suggests that Pictish culture spread to Orkney from Scotland primarily via cultural diffusion rather than direct population movement or inter-marriage. We detect strong continuity between local Iron Age and present-day peoples in

Orkney but less pronounced affinity between early medieval and modern people in eastern Scotland. More ancient genomes from the Iron Age and early medieval periods in the UK are necessary to illuminate these relationships further, combined with analyses of lifetime mobility using complementary approaches (e.g., isotope analysis). On a more local level, our mtDNA analysis of individuals interred at Lundin Links is arguably inconsistent with matrilocality. This finding challenges the older hypothesis that Pictish succession was based on a matrilineal system, assuming that wider Pictish society was organised in such a manner.

## Materials and methods

### DNA extraction, library preparation and sequencing

All aDNA work was carried out in dedicated facilities at Stockholm University. The samples were decontaminated by removing the outer surfaces via abrasion using a Dremel drill after a thorough cleaning in 1% sodium hypochlorite, followed by wiping with molecular biology-grade water and ethanol, and UV irradiation at 254 nm in a crosslinker (Ultra-Violet Products Ltd., Cambridge, UK) for 10 min each side at a distance <10 cm. Approximately 100–200 mg of bone or tooth (dentine) powder was extracted from each specimen using a Dremel drill at the lowest possible rotation-per-minute (5000 rpm).

DNA was extracted using 1 mL extraction buffer consisting of 0.45 mL EDTA (pH8), 1 M urea, and 10 uL of 10 mg/ml proteinase K. The mixture was incubated overnight (~18 hrs) at 37˚C and purified on Qiagen MinElute columns following the manufacturer's recommendation, but with an additional wash step. DNA was eluted in a 63 uL Qiagen Elution buffer. Illumina-compatible sequencing libraries were constructed following Meyer and Kircher (2010) [67] as outlined in Rodríguez-Varela et al. (2017) [68] and sequenced on an Illumina HiSeq2000 platform. Non damage repair libraries from BAL003-b1e1l1 and LUN004-b1e1l1 were sequenced over three and one lanes, respectively. BAL003 and LUN004 libraries were generated using enzymatic damage repair [35], from the same extracts as BAL003-b1e1l1 and LUN004-b1e1l1, respectively, and sequenced over 5 lanes for LUN004 and 15 lanes for BAL003. The libraries for the remaining samples were not damage-repaired. LUN001 and LUN003 libraries were sequenced over seven lanes each. LUN005, LUN006, LUN009 libraries were sequenced over one lane.

### Sequence processing and alignment

We discarded reads with indexes showing at least one mismatch. Read pairs were merged and adapter sequence removed using Adapter Removal v2.1.7 [69], with a minimum overlap of 11 bp and summing base qualities in overlapping regions. Merged read pairs were mapped as single-end reads to the human reference genome build 37 with decoy sequences (hs37d5) using BWA aln v0.7.8 [70] with the non-default parameters -n 0.01 (maximum edit distance) and -o 2 (maximum number of gap opens), allowing more mismatches and indels, and disabled seeding with -l 16500 as in Lazaridis et al. (2014) [1] and Skoglund et al. (2014) [71]. We collapsed duplicate reads having identical start and end coordinates into consensus reads using FilterUniqueSAMCons.py [72]. Finally, we filtered the alignment so that only reads longer than 35 bp, with mapping quality >30, not containing indels, and with more than 90% matches with the reference were retained. We merged libraries sequenced over several lanes using SAMTOOLS v1.9 [70]. Summary statistics of the obtained reads are presented in S1 Table.

### Identification of authentic aDNA molecules and contaminant DNA

We used MapDamage v2.0 [73] to visualise the substitution distribution along the reads and evidence the presence of damaged aDNA molecules. Contamination was estimated using three

different data sources, namely: 1) the mitochondrial genome, 2) X-chromosome contamination in males, and 3) Y-chromosome contamination in females. We estimated present-day mtDNA-based contamination using Schmutzi [74]. For males we used ANGSD [75], which utilises heterozygous calls on the X-chromosome of male samples, expected to be haploid, to estimate contamination. For females, Y-chromosome contamination was assessed by comparing the observed number of reads mapped on the non-pseudo-autosomal region of the Y-chromosome with the expected number if the sample was male. the expected number of Y-chromosome reads is approximated as half the number of reads mapping to the autosomes multiplied by the Y-chromosome fraction of the genome; this assumes the alignment efficiency for the Y-chromosome and autosomes are similar. The Y-chromosome makes up 2% of the genome [76].

## Sex determination

The biological sex of sequenced individuals was determined using the $R_y$ parameters [77]. $R_y$ is the fraction of the Y-chromosome alignments ($n_y$) compared to the total number of reads aligned to the X- and Y-chromosomes ($n_x + n_y$). The 95% confidence interval (CI) was computed as $Ry \pm \frac{1.96 \ x \ Ry \ x(1-Ry)}{(ny+nx)}$. This method determines whether an ancient individual can be determined to be male or female. If the lower CI limit of Ry is >0.077 the individual is assigned as male. If the upper CI limit of Ry is <0.016 the individual is assigned as female.

## Mitochondrial and Y-chromosome haplogroups

We obtained the consensus mitochondrial DNA from endogenous reads, removing the bases with quality <30 (-q 30) using Schmutzi [74]. The mitochondrial haplogroups were assigned using Haplogrep2 [78]. The Y-chromosome haplogroup was obtained using pathPhynder with the parameter -m 'no-filter' [79], based on approximately 120,000 SNPs present in a dataset of Y-chromosomal variation in worldwide present-day and ancient males and the International Society Of Genetic Genealogy (ISOGG, http://www.isogg.org).

## Genomic analysis of pseudo-haploid sequences

The pseudo-haploid genomes of BAL003 and LUN004 were analysed with a set of published pseudo-haploid ancient genomes, the Human Origins dataset and the Simon Genome Diversity Project from the Allen Ancient DNA Resource (https://reich.hms.harvard.edu/) using PCA [80], ADMIXTURE analysis [81], D- and f-statistics [80, 82] and qpAdm [4] (section S1.3 in S1 Text).

## Imputation

Genomes of 33 individuals with a coverage >0.7X from nine geographic regions were imputed using GLIMPSE [83], namely: 1) two early medieval individuals from Scotland dated from the Pictish period (BAL003 and LUN004, this study); 2) three Iron Age (I0789, I0156 and I0160) [17], four Roman British (6DT3, 6DT18, 6DT21 and 6DT22) [14] and eight early medieval individuals from England (I0769, I0773, I0774, I0777, I0157, I0159, I0161 and NO3423) [14, 17]; 3) one Iron Age individual from Norway, 4) six Iron Age individuals from Sweden; 5) three early medieval individuals from Hungary (SZ15, SZ43 and SZ45) [15]; 6) two early medieval individuals from Germany (Alh1 and Alh10) [16]; 7) one Iron Age/early medieval individual from Slovakia (DA119) [84]; 8) one early medieval individual from the Czech Republic (RISE569) [3]; 9) two pre-Christian individuals from Iceland (SBT-A1 and DAV-A8) [85] (section S1.4 in S1 Text and S7 Table).

## Genotype phasing

The EU and UK datasets were phased together with the genomes from Margaryan et al. [9] (S7 Table and section S1.5 in S1 Text) and the re-imputed and newly imputed ancient genomes using BEAGLE 5.2 [86]. We restricted the phasing on the intersections of the genotypes newly imputed in this study and those imputed in Margaryan et al. [9] to prevent sporadic missing genotypes imputation by BEAGLE 5.2 [86]. The window and overlap lengths were set as wider than any chromosome (window length 380 cM and overlap length 190 cM) to maximise the information used for phasing the genomes. The 1000 genomes phase 3 dataset (http://bochet. gcc.biostat.washington.edu/beagle/1000_Genomes_phase3_v5a) and GRCh37 genomic maps (http://bochet.gcc.biostat.washington.edu/beagle/genetic_maps/) provided by BEAGLE were used for phasing. Imputation and genotyping errors can increase phasing errors. However, the BEAGLE phasing algorithm (Hidden Markov Model-based haplotype clustering) improves widely as the sample size increases. The improvement due to the sample size minimises the phasing error from possible genotyping and imputation biases.

## Reference panel

We compiled four datasets (section S1.5 in S1 Text), 1) a set of 1,764 modern individuals from 18 worldwide populations from the 1000 genomes project phase 3 [87], 2) a set of 10,299 modern individuals living in Europe from the EGAD00000000120 International Multiple Sclerosis Genetics Consortium & The Wellcome Trust Case Control Consortium 2 dataset [88], 3) a set of 2,578 modern individuals born in the UK and whose four grandparents were born in the UK within 80 km of each other from the EGAD00010000632 People of the British Isles dataset [11, 53] and 4) a set of 252 ancient European genomes dated from around the Viking Age period [9]. The ancient genomes in this dataset, together with those selected for imputation, were also pseudo-haploidised prior to imputation following the S1.3 in S1 Text method (i.e., to cross-check some results based on the imputed genomes).

## Principal Component Analysis

A PCA was generated using the re-imputed and newly imputed ancient genomes, the ancient genomes from Margaryan et al., (2020) [9] and the modern EU and UK dataset using PLINK v1.9 [89, 90]. SNPs with minor allele frequency (maf) <5% and in linkage disequilibrium (—indep-pairwise 100 10 0.2) were excluded with PLINK v1.9 [89,90]. The PCA was generated on 8,389 individuals and 88,040 SNPs.

A second PCA was generated to increase variability around the ancient British genomes. It was restricted to the modern individuals from Belgium, Denmark, Germany, Norway and the UK, the ancient genomes from the Iron Age and Viking Age individuals from Orkney and re-imputed and the newly imputed ancient genomes in this study from the British Isles, Iceland, Scandinavia, Germany, Slovakia and the Czech Republic following the same method as above. The PCA was generated on 4,914 individuals and 87,518 SNPs.

## ADMIXTURE

Ancestry components were estimated in modern individuals from Belgium, Denmark, Germany, Norway, and ancient genomes from the Iron Age and Viking Age individuals from Orkney and the newly imputed ancient genomes from the British Isles, Scandinavia, Germany, Slovakia and Czech Republic using the program ADMIXTURE v1.2 [81]. Sites with maf <5% and SNPs in linkage disequilibrium (—indep-pairwise 100 10 0.2) were removed with PLINK

v1.9 [89,90], leaving 4,912 individuals and 75,295 SNPs. ADMIXTURE was run with cross-validation (CV) enabled using–cv flag and 50 bootstraps from K = 2 to K = 10.

### D- and F-statistics

Individual or population relatedness was performed using *outgroup-f3* statistics with the qp3Pop function from ADMIXTOOLS 2 R package (https://github.com/uqrmaie1/admixtools). *F3-statistics f3(Yoruba; A, B)* measure allele frequency correlation between populations. When X is an equidistant outgroup to A and B, *outgroup-f3* becomes a genetic drift measure between A and B. The outgroup Yoruba is expected to be equidistant from all tested samples.

To obtain information on individuals or population admixture, we performed outgroup *D-statistics* of the form $D(A, B; C, Yoruba)$ using the qpDstats function from ADMIXTOOLS 2 R package (https://github.com/uqrmaie1/admixtools). A, B and C are either present-day or ancient populations/individuals. A result equal to 0 means that the proposed tree (((A, B), C), Yoruba) is consistent with the data. If D deviates from 0, there are more alleles shared than expected given the proposed tree either between A and C ($D > 0$) or between B and C ($D < 0$).

### Identity-By-Descent and Homozygosity-By-Descent

The identification of IBD and HBD segments was done using RefinedIBD [59]. The window size was set to 3 cM. The minimal size for a segment to be considered shared by IBD or HBD is 1 cM, the same threshold used in Margaryan et al. [9]. We decided to consider segments >1 cM as shared by IBD since 1 cM corresponds to the timespan between the oldest samples from the Iron Age and present-day populations. A common ancestor $n$ generations in the past ($2n$ meiosis) results on average in $100/2n$ cM segment length [91]. Thus, IBD segments longer than 1 cM derive from common ancestors living ~50 generations in the past, or ~1,500 years ago; this assumes an average human generation time of 30 years [92], which is the period roughly including samples from the Iron Age until the present. The total number and total length of shared IBD segments were generated. To avoid sample size bias, we used bootstrapping to compare total number and length of shared IBD over an equal number of individuals when using modern populations with unequal numbers of individuals. We randomly selected an equal number of modern individuals across populations to calculate the total number and length of shared IBD and averaged using 100 replicates with replacement (section S1.6 in S1 Text).

Additionally, we generated interpolated frequency maps of the total number of shared IBD between modern populations and ancient individuals/populations from the UK with QGIS v3.14.1 [93] using distance coefficient P = 2 and pixel size = 0.01. We used the county town as a proxy for the county geographic coordinate.

### ChromoPainter and FineSTRUCTURE

ChromoPainter is a tool for finding the closest haplotypes in sequence data where each individual is painted as a combination of all other genomes using the Li-Stephen Model [94]. ChromoPainter paints the genome of each 'recipient' using all the remaining individuals as 'donors'. We used ChromoPainter v2.9.0 [42] to paint all of the ancient individuals (n = 284). We realised the PCAs based on the ChromoPainter co-ancestry matrix using FineSTRUCTURE GUI.

FineSTRUCTURE v2.9.0 [42] was then used to perform population assignment based on the ChromoPainter coancestry matrix. In brief, FineSTRUCTURE is similar in concept to ADMIXTURE but accesses a large number of SNPs and linkage disequilibrium (haplotype) information. FineSTRUCTURE uses an MCMC approach to partition the dataset into K

groups with indistinguishable genetic ancestry, interpreted as individual populations. The program was run using 100,000 iterations and 100,000 burn-in.

## Supporting information

**S1 Table. DNA sample information from eight Pictish genomes in this study.**
(XLSX)

**S2 Table. Sample misassignment estimate in LUN001 and LUN003 using qpAdm.**
(XLSX)

**S3 Table. Sample misassignment rate in LUN001 and LUN003 mtDNA reads.**
(XLSX)

**S4 Table. Calibrated $^{14}$C radiocarbon dates.**
(XLSX)

**S5 Table. Mitochondrial DNA mutations and haplogroup attribution.**
(XLSX)

**S6 Table. Y-chromosome mutations on LUN004.**
(XLSX)

**S7 Table. Comparison datasets for the analysis of the pseudo-haploid genomes and imputed genomes from Margaryan et al. [9].**
(XLSX)

**S8 Table. Shared genetic drift between the BAL003, LUN004 and modern populations as f3(BAL003/LUN004, modern;Mbuti).**
(XLSX)

**S9 Table. D(Iron Age/Roman English, Iron Age/Medieval European; BAL003/LUN004, Mbuti).**
(XLSX)

**S10 Table. D(Iron Age/Roman English, BAL003/LUN004; modern and ancient populations, Mbuti).**
(XLSX)

**S11 Table. Coancestry matrix obtained with ChromoPainter for the analysis of 284 ancient European genomes dated from the Iron Age to medieval period.**
(XLSX)

**S12 Table. Genotypes associated with lactase persistence and skin, eyes and hair pigmentation with subsequent prediction performance metrics from HIrisPlex-S.**
(CSV)

**S1 Fig. Plan of cists at Lundin Links.** The plan was published in Greig et al. [36]. The samples yielding DNA are in red.
(TIF)

**S2 Fig. Calibrated $^{14}$C radiocarbon dates.** Radiocarbon determinations calibrated using OxCal. v4.4 [95] and the IntCal20 atmospheric curve [96].
(PNG)

**S3 Fig. Deamination pattern along the reads.**
(TIF)

**S4 Fig. Biological sex.** We determined the biological sex based on Y-chromosome alignments (ny) compared to the total number of reads of the X and Y-chromosomes (nx + ny). The 95% confidence interval (CI) was computed as Ry ±(1.96 x Ry x (1 –Ry))/((ny + nx)).
(TIF)

**S5 Fig. Genetic distance between BAL003, LUN004, Iron Age and Roman individuals from England and Iron and Viking Age individuals from Orkney (excluding VK204 and VK205) measured using outgroup-f3 statistics.**
(TIF)

**S6 Fig. Cross-validation error of the ADMIXTURE analyses.** The cross-validation was done using the—cv option of admixture runs for K = 1 to K = 10 with 4,914 individuals and 87,518 SNPs using 50 bootstraps (-B 50).
(TIF)

**S7 Fig. ADMIXTURE Model-based clustering analysis of 4,914 modern and ancient Northern and Central Europeans from the Iron Age, Early Medieval period and present-day (K = 2 to K = 10).**
(TIF)

**S8 Fig. Principal Component Analysis of the 8,389 present-day and ancient individuals from this study.** SNPs with maf <5% were removed and pruned (88,040 SNPs remained). PC3 is impacted by allele frequency bias differentiating the data imputed in Margaryan et al. [9] and the genomes newly imputed in this study.
(TIF)

**S9 Fig. Correlation between coverage and PCs coordinates from the PCA presented in S8 Fig.** The genomes newly imputed in this study are in blue and the genomes imputed in Margaryan et al. [9] are in pink.
(TIFF)

**S10 Fig. Genetic affinity of BAL003 and LUN004 compared to western Eurasians using PCA.** Pseudo-haploid genotypes from ancient samples are projected onto the first two principal components defined by 1,056 present-day West Eurasians from the 'HO' dataset (S). VA, Viking Age.
(TIFF)

**S11 Fig. Cross-validation error of the ADMIXTURE analysis.** The cross-validation was done using the—cv option of admixture runs for K = 3 to K = 20 with 3,591 individuals and 85,655 markers.
(TIF)

**S12 Fig. ADMIXTURE ancestry component (K = 15) of ancient shotgun and present-day worldwide individuals.** Full displays of the ADMIXTURE analysis are in S13–S15 Figs.
(TIFF)

**S13 Fig. Model-based clustering analysis of 3,594 individuals (K = 3 to K = 20).** Only modern individuals are represented.
(TIFF)

**S14 Fig. Model-based clustering analysis of 3,594 individuals (K = 3 to K = 20).** Only ancient individuals from western Eurasia are represented.
(TIFF)

**S15 Fig. Model-based clustering analysis of 3,594 individuals (K = 3 to K = 20).** Only ancient individuals from Eastern Eurasia are represented.
(TIFF)

**S16 Fig. Testing for symmetry between the pseudo-haploid BAL003 and LUN004 genomes and the Iron Age or Roman period individuals from England relative to other ancient and modern populations.** We tested for symmetry as D(Pict, ancient England; X, Mbuti) and plotted the resulting Z-scores. Details on the test results and sample size are in S10 Table.
(TIF)

**S17 Fig. Testing for continuity between European Iron Age/medieval period and BAL003 and LUN004 pseudo-haploid genomes using qpAdm in a one-way admixture model.**
(TIF)

**S18 Fig. Modelling Bronze Age, Iron Age and medieval European populations as a mixture of Western Hunters-Gatherers (WHG), Anatolian Neolithic (Anatolia_N) and Bronze Age Yamnaya using qpAdm on pseudo-haploid genomes.** Empty bars are target populations for which the three-way models' proportion was impossible to estimate using qpAdm. Vik_97002. SG, vik_97026.SG, vik_urm045.SG and vik_urm161.SG are individuals buried in a Viking context in Sigtuna, Sweden but likely migrants of diverse origins. G, generation.
(TIF)

**S19 Fig. Shared Identity-By-Descent (IBD) segments between and within present-day European populations.** A) Total length of shared IBD segments >1 cM, B) total number of shared IBD segments >1 cM, C) total number of shared IBD segments >4 cM and D) total number of shared IBD segments >6 cM. The number corresponds to the mean of 100 bootstraps drawing 44 random individuals per population.
(TIF)

**S20 Fig. Shared Identity-By-Descent (IBD) segments between ancient genomes and modern European populations.** The ancient genomes are the newly imputed genomes and the six ancient Orcadians from Margaryan et al. [9]. A) Total length of shared IBD segments >1 cM, B) total number of shared IBD segments >1 cM, C) total number of shared IBD segments >4 cM and D) total number of shared IBD segments >6 cM. The number corresponds to the mean of 100 bootstraps drawing 44 random individuals per European population.
(TIF)

**S21 Fig. Shared Identity-By-Descent (IBD) segments between ancient individuals from Britain.** A) Total length of shared IBD segments >1 cM, B) total number of shared IBD segments >1 cM, C) total number of shared IBD segments >4 cM and D) total number of shared IBD segments >6 cM. IA, Iron Age; VA, Viking Age; Emedieval, early medieval.
(TIF)

**S22 Fig. Number of Identity-By-Descent (IBD) segments shared between present-day and ancient genomes from Britain.**
(TIF)

**S23 Fig. Distribution of Identity-By-Descent (IBD) length as a function of the time difference between pairs of samples.**
(TIF)

**S24 Fig. Principal Component Analysis (PCA) of the 8,044 present-day Europeans individuals.** The pseudo-haploid ancient genomes (S7 Table) were projected. This analysis replicates

S8 Fig but only using modern genomes for the PCs computation and projected pseudo-haploid ancient genomes. SNPs with maf <5% were removed and pruned (88,369 SNPs remained). (TIFF)

**S25 Fig. Principal Component Analysis (PCA) of the 4,863 present-day western Europeans individuals.** The pseudo-haploid ancient genomes (S7 Table) were projected. This analysis replicates Fig 2A, but only using modern genomes for the PCs computation and projected pseudo-haploid ancient genomes. SNPs with maf <5% were removed and pruned (88,369 SNPs remained). (TIFF)

**S26 Fig. Distribution of Homozygosity-By-Descent (HBD) across ancient individuals from Britain.** (TIF)

**S27 Fig. PCA based on ChromoPainter coancestry matrix and FineSTRUCTURE clustering generated for 284 ancient European genomes dated to the Iron Age and medieval period.** (TIF)

**S1 Text. Supplementary information.** (DOCX)

# Acknowledgments

We thank Gareth Weedall at Liverpool John Moores University for his valuable help with editing the manuscript, Jeanette Pearson at Inverness Museum & Art Gallery for facilitating sampling of Balintore material, Jean-Jacques Hublin, Mike Richards (at the time both MPI-EVA), and the Max Planck Society for laboratory access and resources in kind and Orsolya Czére at University of Aberdeen for support with C:N determination.

# Author Contributions

**Conceptualization:** Adeline Morez, Kate Britton, Gordon Noble, Torsten Günther, Anders Götherström, Ricardo Rodríguez-Varela, Nicholas J. Evans, Linus Girdland-Flink.

**Data curation:** Adeline Morez, Kate Britton, Linus Girdland-Flink.

**Formal analysis:** Adeline Morez, Gordon Noble, Rui Martiniano.

**Funding acquisition:** Anders Götherström, Linus Girdland-Flink.

**Investigation:** Adeline Morez, Kate Britton, Ricardo Rodríguez-Varela, Natalija Kashuba, Nicholas J. Evans, Linus Girdland-Flink.

**Methodology:** Adeline Morez, Torsten Günther, Sahra Talamo, Linus Girdland-Flink.

**Project administration:** Adeline Morez, Linus Girdland-Flink.

**Resources:** Kate Britton, Gordon Noble, Anders Götherström, Rui Martiniano, Sahra Talamo, Christina Donald, Linus Girdland-Flink.

**Software:** Adeline Morez, Rui Martiniano.

**Supervision:** Anders Götherström, Linus Girdland-Flink.

**Validation:** Adeline Morez.

**Visualization:** Adeline Morez.

**Writing – original draft:** Adeline Morez, Joel D. Irish, Linus Girdland-Flink.

**Writing – review & editing:** Adeline Morez, Kate Britton, Gordon Noble, Torsten Günther, Ricardo Rodríguez-Varela, Natalija Kashuba, Rui Martiniano, Sahra Talamo, Nicholas J. Evans, Joel D. Irish, Christina Donald, Linus Girdland-Flink.

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
