## [Decision Letter · Decision Letter 0]

22 Sep 2022

Dear Dr Morez,

Thank you very much for submitting your Research Article entitled 'Imputed genomes and haplotype-based analyses of the Picts of early medieval Scotland reveal fine-scale relatedness between Iron Age, early medieval and the modern people of the UK.' to PLOS Genetics.

The manuscript was fully evaluated at the editorial level and by independent peer reviewers. The reviewers appreciated the attention to an important problem, but raised some substantial concerns about the current manuscript. Based on the reviews, we will not be able to accept this version of the manuscript, but we would be willing to review a much-revised version. We cannot, of course, promise publication at that time. 

Should you decide to revise the manuscript for further consideration here, your revisions should address the specific points made by each reviewer. In particular, Reviewers 2 and 3 suggested possibilities that can strengthen the analyses, while Reviewer 1 pointed to a number of caveats that may confound interpretations. We encourage you to incorporate these comments in your revisions. We will also require a detailed list of your responses to the review comments and a description of the changes you have made in the manuscript.

If you decide to revise the manuscript for further consideration at PLOS Genetics, please aim to resubmit within the next 60 days, unless it will take extra time to address the concerns of the reviewers, in which case we would appreciate an expected resubmission date by email to plosgenetics@plos.org.

[LINK]

We are sorry that we cannot be more positive about your manuscript at this stage. Please do not hesitate to contact us if you have any concerns or questions.

Yours sincerely,

Jonathan Marchini

Academic Editor

PLOS Genetics

Hua Tang

Section Editor

PLOS Genetics

Reviewer's Responses to Questions

**Comments to the Authors:**

Reviewer #1: Morez et al. use ancient DNA to investigate the genetics of the ancient ‘Picts’ – groups who inhabited aspects of present-day Scotland in the early medieval period. The origins of these groups of people and what language they spoke have been longstanding questions in the history of Britain, with multiple theories deriving from linguistic comparisons as well as histories written by contemporary and later historians. Even though Britain is one of the most studied areas of the world when it comes to ancient DNA, so far there has been no head-on attempt to investigate what ancient DNA can tell us about the Picts.

Morez et al. are the first to directly investigate Pictish genetics by, in the first instance producing one medium and one high coverage ancient genome from disparate parts of Pictish territory. However, the main novelty and power of their paper is in applying imputation methods to previously published data to explore fine-scale genetic affinities of Iron Age and early medieval populations of Britain in a way that has not been possible previously. They therefore produce a whole range of insights not only about the Picts, but also other ancient populations of Britain and their relationship to some of the populations who inhabit the same regions today. Morez et al. also generate mitochondrial genome data from a further 7 Pictish samples from the Lundin Links site to investigate hypotheses established from historical sources that Pictish groups were matriarchal. The diversity of maternal lineages they discover at this cemetery suggests that burial was not based on matrilineal descent. However, the authors are careful to discuss a whole range of caveats to any conclusions about social organisation in Pictish societies from a small number of samples from a single site. The paper is well-written and clear, uses appropriate methods in novel ways and includes some excellent figures. Therefore I believe is well worthy of publication. I just had a few minor comments that I think the authors could consider:

Page 2, Line 22: ‘The origins and ancestry of the Picts of early medieval Scotland (ca. AD 300-900) has been traditionally seen as a problem, prompted in part by exotic medieval origin myths, their enigmatic symbols and inscriptions, and the meagre textual evidence.’

The wording of this sentence comes across as slightly odd to me. ‘Problem’ in this context means something potentially harmful rather than a question to be resolved. I think the authors could consider rephrasing, perhaps something like, ‘There are longstanding questions about the origins and ancestry of the Picts of early medieval Scotland (ca. AD 300-900), prompted…’

Page 3, Line 59: ‘The British Isles witnessed a complex cultural turnover from the Iron Age to the early medieval period. The Romans occupied part of Britain to southern Scotland from AD 43 to ca. AD 410; however, this occupation resulted in little detectable gene flow from mainland Europe (14).’

The authors use the term ‘British Isles’ throughout. Are the authors using ‘British Isles’ to refer to the islands associated with Britain and Ireland or just Britain? While ‘British Isles’ is primarily a geographic term and is strictly accurate when discussing Britain and Ireland (along with their smaller islands together), it is still considered by some to be politically loaded, particularly in the Republic of Ireland where they (understandably) take issue being referred to as ‘British’. If the authors are using this term just to refer to Britain and its associated islands, then ‘British Isles’ is a bit confusing because of this history of use – If the authors are using it this way they should probably define what they mean by it. It is up to the authors whether they want to adjust this, but alternatives which are less politically sensitive include ‘British and Irish Isles’, or just ‘Britain and Ireland’, with some definition specifying that this refers to both or either of these countries and associated smaller islands.

Page 3, Line 61: ‘The Romans occupied part of Britain to southern Scotland from AD 43 to ca. AD 410; however, this occupation resulted in little detectable gene flow from mainland Europe (14).’

I think this is quite a bold statement from the authors deriving from a single study of a single Roman site, particularly as later genetic shifts might have distorted or erased any pre-early medieval gene flow. Also, the authors themselves later find evidence that this may not be strictly true in their reanalysis of the same site. Perhaps ‘…however, the single ancient DNA study of Roman Britain suggests this occupation…’

Page 4, Line 84: ‘In the modern era perceptions of Pictish origins have varied, often according to cultural and political biases, with the Picts and their languages regarded as Germanic, Gaelic, Brittonic, Basque, and Illyrian, among other theories. In the 1950s Jackson influentially argued that the Picts spoke a non-Indo-European language and a Celtic language akin to ancient Gaulish (23,24). The current consensus is that they spoke a Celtic language closest to that of neighbouring Britons from which Cornish, Welsh, and Breton derive (25–27). However, some still argue from undeciphered inscriptions and other words that some Picts spoke an otherwise unknown language, presumably derived from a pre-Celtic population (28,29). Thus, the question remains of whether the Picts were somehow fundamentally different from their neighbours.’

I may have missed it, but I think the authors could consider mentioning a further complication, particularly when it comes to links between language and genetics - that the Patterson et al. (2021) paper found that genetic change in southern Britain produced by Late Bronze Age migrations which may have introduced Celtic languages or certain forms of Celtic to these areas, was not so influential in northern Britain.

Page 7, Line 147: ‘but without additional resolution to determine whether this sample carries the R1b-P312/S116 haplogroup introduced to Britain by Bell Beaker peoples during the Chalcolithic, alongside Yamnaya-related ancestry (7).’

I think it would be useful for the authors to consider whether it is better to refer to steppe-related or Western Steppe Herder (WSH) ancestry rather than Yamnaya-related, as these terms are not culturaly specific and there has been some recent evidence that this ancestry may not be so much derived from Yamnaya groups (although I suppose ‘Yamnaya-related’ is still technically accurate).

Page 8, Line 158: ‘The PCA shows that the ancient individuals from Britain broadly fit within present-day diversity (Fig 159 2A).’

I think this sentence is a bit unclear – present-day diversity of where? Also, I think the authors (and indeed, all geneticists to some extent) could be more careful to be precise when discussing present-day populations. The present-day samples referred to here do not represent the diversity of present-day people in Britain but of people whose recent ancestors predominantly came from Britain. I appreciate this kind of thing is understood between geneticists, but increasingly there will be many non-specialists interested in this paper who sometimes get the wrong idea of what geneticists mean when referring to present-day populations. I don’t think the authors need to define what they mean by present-day groups every time they are mentioned, perhaps just once on their first mention.

Page 9, Line 194: ‘This implies that we cannot consider individuals from Pictland a homogenous genetic group but instead a complex mixture of contemporary genetic ancestries.’

I think ‘implies’ might be the wrong word here as the authors make this observation directly. ‘Suggests’ or ‘indicates’ might be better, although I think beginning the sentence from ‘We’ would also be justifiable and make a stronger statement.

Page 10, Line 213: ‘In fact, recent research show that Bronze Age populations in Orkney were differentiated from their counterparts on mainland Britain due to retention of male Neolithic ancestry (Y-chromosomal haplogroup I2), while the R1b haplogroup associated with Bell Beaker expansion largely replaced the I2 haplogroups in the rest of Britain (13), implying that local ancestry may have persisted also into the Iron Age and early medieval period’

This is a very interesting idea. However, if there had been significant retention of ancestry from Neolithic groups in Orkney into the Bronze Age beyond paternal lineages, wouldn’t we see this in the whole ancestry of Bronze Age Orcadians? From what I understand, Dulias et al. (2022) found the ancestry turnover in Orkney was just as large as in mainland Scotland. Or are the authors speculating that there may have been remnant local pockets of Neolithic-derived ancestry amongst unsampled communities in Bronze Age Orkney? If so, I think this could be clearer. If not, would this then suggest significant genetic drift in Orkney through the Bronze Age is more likely to be responsible? In fact, Dulias et al. (2022) observed such Bronze Age drift directly.

Page 10, Line 220: ‘This is consistent with our results that show a high proportion of shared IBD segments among modern Orcadians (>1 to >6 cM, S19 Fig), meaning they share a high proportion of recent common ancestors relative to most modern European populations, typical of small or genetically isolated populations.’

This is an excellent, brief explainer of what IBD effectively means in practice, nice work, very useful!

Page 11, Line 243: ‘This pattern suggests that Anglo-Saxon ancestry expanded out of south-eastern England followed by admixture with local populations, which is a scenario consistent with previous research (11,14,17,42,43).’

I think the authors could consider whether discussion of ‘Anglo-Saxon ancestry’ is appropriate here. ‘Anglo-Saxon’ is a cultural grouping and one which as far as we know seems to encompass people with ancestries derived both from northern continental Europe and Roman Britain (Schiffels et al. 2016). Therefore discussing ‘Anglo-Saxon ancestry’ seems imprecise to me as well as having some political potency. Something along the lines of ‘This pattern suggests ancestry from northern continental Europe associated with Saxon migrations expanded out of south-eastern Britain…’ might reflect the situation more precisely without making such a direct link between culture and DNA.

Page 12, Line 262: ‘Alternatively, the ancestors of BAL003 and LUN004 share more IBD segments with present-day people from western Scotland, Wales, and Northern Ireland because they (or their direct ancestors) migrated from these regions but did not contribute substantially to later generations via admixture with local groups in eastern Scotland.’

Given that some of the historical sources make clear a connection between the Picts and Ireland (as the authors discuss briefly I their introduction), the primary thing that feels to be missing from this paper is a fuller comparison of Picts against populations from Ireland. This especially true given the author’s interesting finding that BAL003 and LUN004 have affinities to present-day people from northern Ireland and other place in Britain (west of Wales, Argyll) that have been influenced by Irish migration in more recent times. The authors do hint at this possibility in posing a scenario where both Pictish individuals were recent migrants or descendants of migrants from further west, but couldn’t this be explored in more detail? I appreciate that there is little relevant ancient data from Ireland, but couldn’t the present-day Irish data from the Gilbert et al. (2017) paper help here? For instance if BAL003 and LUN004 share ibd segments with mainly northern Ireland, this may suggest that sharing between Pictish samples and Ireland might be down to more recent interactions between northern Ireland and southwestern Scotland. If the Pictish samples shared affinities with people on the island of Ireland more generally, this may be more suggestive of an ancient Irish influence. Alternatively, are the Bronze Age samples from Rathlin island too temporally distant to be of use here? Could comparison of the authors’ Pictish genomes against Bronze Age samples from Ireland (with an assumption of some level of continuity into the Iron Age) and Iron Age samples from Scotland help to tease out any ancient Irish-specific genetic influence?

Page 13, Line 286: ‘Therefore, the marked genetic differentiation between the Orkney and mainland Britain is not only a result of Scandinavian admixture, as previously hypothesised (11,42,50–53) but also pronounced genetic continuity that persisted for at least 2,000 years.’

Typo - ‘the Orkney’

Page 13, Line 297: ‘6DT3 also share relatively more IBD segments >1 cM with the present-day population from Scandinavia, Belgium and the UK (Fig 3), suggesting that Scandinavian-like ancestry could have spread to the British Isles before the Anglo-Saxon period.’

As the authors are discussing Britain, rather than just England, I think they could consider whether it would be better to stick to ‘early medieval’ rather than use ‘the Anglo-Saxon period’. The authors tend to use ‘early medieval’ period throughout the paper and so using it here would be more consistent.

Page 14, Line 300 ‘Seven mtDNA genomes were retrieved at Lundin Links, which allows us to answer questions about the Pictish social organisation reflected in the individuals interred at the site.’

I’m not sure if the authors ‘answer’ questions about Pictish social organisation here – maybe ‘…address questions…’?

I am impressed by the way the authors qualify their statements about inferring social organisation from ancient genomic data and go into some of the complexities that make it difficult to be sure of their interpretations. Having said that, I do feel one thing that is missing or at least not as prominent as it could be is that the authors (as in other archaeogenetics papers) seem to take it for granted that cemeteries include people who cohabited in life and so provide an insight into who stayed and who left a particular settlement. For instance, there could be a situation where bodies are returned to their natal communities despite having lived their lives within the communities they married into. I think it is totally reasonable to discuss social organisation with this assumption in mind but I think the assumption has to be made explicit to show what interpretive steps are being taken.

Page 14, Line 305: ‘It is worth noting that the two individuals retrieved from the horned

cairns complex individuals (S1 Fig) show evidence of familial links based on skeletal morphology

(LUN001 and LUN009) (35).’

I think it would be useful here for the authors to give some brief detail on what skeletal traits suggested a family link just to make it easier for a reader to assess the strength of this evidence.

Page 14, Line 314: ‘Thus, this is unlikely that the community at Lundin Links followed a matrilineal inheritance system, which challenges older arguments for matrilineal succession among Pictish rulers (59).’

As I’ve discussed above, I think an important caveat to the authors’ statement is ‘Insofar as funerary treatment can tell us about social organisation in life…’.

Page 15, Line 343: ‘On a more local level, our mtDNA analysis of individuals interred at Lundin Links is inconsistent with matrilocality. This finding argues against the older hypothesis that Pictish succession was based on a matrilineal system, assuming that wider Pictish society was organised in such manner.’

Given the limitations to this interpretation the authors themselves cover in their discussion, I think these sentences are a bit strong and require some qualification – perhaps a ‘Tentatively…’ at the beginning of the second sentence. Also I’m not sure if the finding ‘argues’ against the hypotheses, maybe ‘goes against’ would be a better fit. There may be in ‘a’ missing from ‘such a manner’.

Supplementary Information:

‘We observed an increase of Anatolian Neolithic ancestry in the Iron Age individual buried in southern England (13.4% ± 4.7 SE) , compared to an Iron Age individual buried in northern England (6.60% ± 17.5 SE) (52)’

Has the second of these figures percentages/standard errors been switched?

‘Evidence for European ancestry in a Romano-British individual from Duffield terrace, Yorkshire.’

I think the name of the site is ‘Driffield Terrace’.

Overall though, an excellent paper, which I thoroughly enjoyed reading. A lot of hard work has been put into this and it shows.

References

Dulias, K., Foody, M.G.B., Justeau, P., Silva, M., Martiniano, R., Oteo-García, G., Fichera, A., Rodrigues, S., Gandini, F., Meynert, A. and Donnelly, K., 2022. Ancient DNA at the edge of the world: Continental immigration and the persistence of Neolithic male lineages in Bronze Age Orkney. Proceedings of the National Academy of Sciences, 119(8), p.e2108001119.

Gilbert, E., O’Reilly, S., Merrigan, M., McGettigan, D., Molloy, A.M., Brody, L.C., Bodmer, W., Hutnik, K., Ennis, S., Lawson, D.J. and Wilson, J.F., 2017. The Irish DNA Atlas: Revealing fine-scale population structure and history within Ireland. Scientific reports, 7(1), pp.1-11.

Patterson, N., Isakov, M., Booth, T., Büster, L., Fischer, C.E., Olalde, I., Ringbauer, H., Akbari, A., Cheronet, O., Bleasdale, M. and Adamski, N., 2022. Large-scale migration into Britain during the Middle to Late Bronze Age. Nature, 601(7894), pp.588-594

Schiffels, S., Haak, W., Paajanen, P., Llamas, B., Popescu, E., Loe, L., Clarke, R., Lyons, A., Mortimer, R., Sayer, D. and Tyler-Smith, C., 2016. Iron age and Anglo-Saxon genomes from East England reveal British migration history. Nature communications, 7(1), pp.1-9.

Reviewer #2: This paper presents newly generated ancient DNA data, from two sites dated to the 4-6th century located in Northern and Southern present-day Scotland. They are considered "Pictish" sites based on their age and location. The new data consists of two whole-genome individuals sequenced (one intermediate and one for aDNA exceptionally high coverage individual with 2.4

and 16.5x average coverage, respectively), and six additional mitogenomes.

The authors apply a high-resolution population genetic analysis to the two individuals with autosomal data, including PCAs based on many present-day Northern and Western Europeans, as well as haplotype sharing analysis, which is cutting edge in aDNA analysis. They find that the two whole genomes cluster close to contemporaneous Northern British samples, which is not a major surprise. There is variability between the two samples and they both cluster seemingly with "Western" British samples (i.e. Walisians and coastal Norther Islanders), even closer than with present-day Eastern Scotlanders which would be closer to the region of the Pictish samples. There is some interesting speculation on why that may be, including a hypothesis that Eastern Scotland received immigration later on. The patterns hold both for classical allele frequency-based and haplotype-sharing methods.

The authors also make a claim that the seven different mtDNA lineages from one Archeological site in Southern Scotland argue against a matrilineal system, a long-standing hypothesis for Pictish society (based on vague textual records).

Regarding the significance of the study, it is notably the first genome-wide data from the Picts. However, as the authors point out themselves, it is not clear how general the results are. Two samples, from two geographically far apart sites, do not allow one to make a conclusion about the general populace of all of Scotland, as there could be population genetic variation, both between and within sites. Nowadays it is a very low sample size, most aDNA studies include dozens or even hundreds of individuals, in particular from more recent times where the number of potential aDNA samples is abundant.

Considering these caveats of low sample size, it is especially striking that two individuals with seemingly well-conserved genome-wide DNA data have been excluded from any autosomal analysis. The reason is suspected cross-contamination from native Americans due to a technical artifact called index hopping (see point A1 below). So the sample size could have been larger, with reasonable monetary effort. That exclusion also raises alarm bells, in terms of data quality and also in terms of potentially biased exclusion of data (see point A1 below).

The population analysis is mostly sound, but there are several oddities in the analysis (see points below). The imputation and IBD part is seemingly cutting edge - but it is effectively a "black box" approach, where algorithms developed for high-quality modern DNA data are applied to aDNA, with very limited verification of the tools and testing for technical covariates (coverage, ancestry, etc.). Thus, it is questionable how well this study can serve as a blueprint for other work - however, I understand that the overall goal here is to screen for qualitative patterns.

Overall, the article is well structured and easy to follow, and the analysis and the interpretation are mostly laid out clearly. The literature and previous studies are generally cited well, at least from an ancient DNA viewpoint (which is my expertise). Previous work on shared segments could be cited more comprehensively, e.g. about relationships between population sizes and segment sharing or expected geographical patterns - to aid discussion of the results.

Below please find detailed comments, first summarize critical comments (A), then major points (B), and then minor suggestions (C) which are not-make-or-break but the authors hopefully find useful.

A: Critical Points (which can be make-or-break pitfalls):

A1) Exclusion of genome-wide data from two samples (LUN001 and LUN003) from any analysis. According do the SI text, LUN003 actually does not show any direct significant evidence of contamination, and the Y-based one for both samples (1 and 3%) is well within the standard limits of aDNA contamination estimates (<10%). The f-statistics-based admixture estimate is highly problematic, as Native American samples are enriched for an ancestry component also high in Steppe-ancestries (e.g. Karitiana are often the top attraction in f-statistics when comparing Steppe ancestries to Neolithic Europeans). In case the two samples are simply higher in Steppe ancestry than the other samples, this admixture test would give significant results, as the assumptions of cladality are broken. A simpler and more robust method would be e.g. a two-way qpADM model with the two putative contaminants as one source, several plausible European ancient individuals as the second source, and other Native Americans as one outgroup.

But given the e.g. low Y chromosome contamination at least showing some basic population genetic analysis (such as PCA) in Supplementary Analysis is clearly warranted. Otherwise, some readers might suspect data has been omitted because it "does not fit or would complicate the story".

Ideally, of course, these data would be actually generated again to be sure - some in the aDNA field might even argue that there is a moral duty to make the most out of the data obtained through destructive sampling.

A2) In Table 1, the mtDNA-contamination estimates and CIs are reported in steps of "1.00%". Is that extreme coarseness the consequence of unwanted rounding?

A3) The Haplotype Analysis consists of "black box" approaches, by applying methods developed for modern DNA data without any evaluation of the results. Some potential pitfalls would need investigation in case the methodology should serve as a blueprint for other studies. IBD calling is a highly challenging task even in modern genomes (see e.g. Ralph & Coop 2013, with extensive testing across IBD calling that confirmed rather large IBD detection errors). Of course, the goal of this paper is to screen for broad patterns, but these potential technical pitfalls should be at least acknowledged in the text. As one example, IBD segments of length 1 cm are extremely hard to detect in present-day data - with little power (<<50%) and large false positive rates (see e.g. Seidman et al, 2020) - that plausibly differ between ancestral backgrounds. Some discussion of these points would be certainly in order.

One possible validation is to compare ROH to any ROH estimates based on methods developed, calibrated, and tested on aDNA data (e.g. hapROH, Ringbauer et al 2021). Here you seem to find no ROH>4cM in the two new ancient samples - which is a finding in itself (of relatively large population sizes) - but, if confirmed with hapROH, you could compare on other ancient samples with longer ROH.

A4) Abstract: "Analysis of mitochondrial DNA diversity at the Pictish cemetery of Lundin Links (n = 7) reveals the absence of female endogamy"

The sample size is very small, and the sample is potentially spread out over 130 years, with even non-overlapping RC dates. Without having any "family" or "pedigree" results it is plausible that the 7 individuals are from different families. Generally, mtDNA diversity is very high in most populations - and it is not clear whether a few generations of "female endogamy" can reduce mtDNA diversity (it's hard for women to have a large number of daughters). Thus, I believe this claim is not strongly supported and the language should be tuned down also in the abstract.

B: Major Points:

B1) A PCA built including imputed ancient genomes has the potential for batch effects (e.g. discussed for PC3 in the SI). Given the vast number of present-day genomes, a much more robust way to do this is to construct the PCA based on modern genomes only, and then project the ancient pseudo-haploid data onto it, using a shrinkage correction (as available in the SMARTPCA software used here). This is the standard in the aDNA field, and e.g. Fig. 2 would very likely look very similar - as the present-day variation dominates the sample in any case. Importantly, it would leave much less room for any bias.

B2) L546-L548, SI: Downsampling for IBD "The total number and total length of shared IBD segments were then calculated. To avoid sample size bias, we randomly selected the same number of modern individuals across populations."

One can simply normalize with the number of screened pairs. That way one can make use of all the modern data and does not need to throw away so many individuals.

B3) L526-L528, SI: "Thus, IBD blocks longer than 1 cM derive from common ancestors living ~50 generations in the past, or ~1,500 years ago"

That is a huge oversimplification. For an insightful primer see e.g. Ralph & Coop 2013.

First, you mean segments ~1cM (longer ones tend to be more recent). Second, other periods contribute to ancestors producing IBD of 1cM too, how much depends on the demographic history. E.g. a bottleneck 3000 years ago still has the potential to contribute very substantially to IBD 1 cm long. Even for a panmictic population, the distribution of recent ancestors contributing to IBD of a given length is very broad (see e.g. Ringbauer et al, 2021, Fig. S14).

B4) Data availability" It seems that only the raw bam files will be shared. However, for overall replicability, it will be extremely helpful if the processed data used in the analysis (both pseudo-haploid and especially the imputed data) would be made publicly available too.

B5) Fig. S1 is very low resolution. The main figures suffer from low resolution and strong pixelations artifacts too, especially the PCA figure (Fig. 2)

B6) One of the two individuals with autosomal data lacks a C14 date (LUN004). Given the relatively low cost of radiocarbon dating (<500 Euro/Dollar), is there any chance to obtain one such date? RC dates can provide critical information and safeguard against unwanted surprises (such as the Neolithic "Picts" mentioned in the text).

B7) The UDG-treated two whole-genome sequences have very little damage (<<5%, even on the end, see Fig. S3), as expected. It is not needed to restrict the analysis to transversions only then (and effectively never done in aDNA practice in these cases). But here this removal of large amounts of data is done for pseudo-haploid analysis and as input for imputation (see below). That throws away large amounts of data without any real need to.

B8) L147-148 "we assigned LUN004 to R1b-L52 (Table 1, S6 Table), but without additional resolution to determine whether this sample carries the R1b-P312/S116" R1b-P312"

A sample with >2x average whole genome (!) coverage should most likely be resolvable. Is it really the case that all of the several equivalent markers (and dozens of potential downstream mutations along each major branch) are not covered at all?

B9) L204-L206: "However, due to allele frequency bias between the two imputed datasets, likely skewing allele frequency-based analyses (S1.4 Text, S8, and S9 Figs), we refrained from calculating D-statistics to investigate this signal further"

Why not calculate the f-statistics on pseudo-haploid data then? f-statistics are extremely robust and their power anyways "tops out" with higher coverage (due to linkage - above 100k pseudo-haploid SNPs usually more SNPs != more power).

C: Minor Points:

C1) L407-411: Imputation

"For non-damaged repaired genomes, genotypes potentially resulting from deamination were set as missing, replacing 0/1 and 1/1 genotypes by ‘./.’ where the reference allele is C or G, and the alternate is T or A or replacing 0/0 and 0/1 genotypes by ‘./.’ where the reference allele is T or A and the alternate is C or G."

Is this not a potential bias, as only derived alleles are being masked out? Imputation will probably not recover all of them, or at least with lower certainty than in cases where there are reference alleles. The way to go would be to call genotype likelihoods with aDNA damage aware likelihoods - e.g. ATLAS or snpAD.

Importantly, given the low remaining damage after the USER enzyme treatment, restricting to only transversion SNPs seems overly conservative in any case (see point B8 above).

C2) L420-L443: You investigate bias on PC3 - showing that there is a coverage-correlated bias for both the Margaryan and your imputation. It would be good to test and mention that on PC1/PC2 there is none - as these PCs are centrally featured in the main text and figures. In any case, you could simply project pseudo-haploid ancient data to avoid these evident interactions of PCA and imputation (see point B1 above).

C3) L181: "To make use of the additional power provided by linkage disequilibrium"

That's an anal technical comment: Haplotype analysis does not need any LD - there can be shared long haplotypes even in a population without any substantial LD. So you really want to say: "the additional power provided by haplotype information"

C4) B3) L565, SI: The "quality control" section first explains some heuristics of the length-cutoff - and then reports that patterns qualitatively agree with present-day IBD patterns. Titling such a section as "quality control" seems ambitious.

References:

- Ralph, Peter, and Graham Coop. "The geography of recent genetic ancestry across Europe." PLoS biology 11.5 (2013): e1001555.

- Seidman, Daniel N., et al. "Rapid, phase-free detection of long identity-by-descent segments enables effective relationship classification." The American Journal of Human Genetics 106.4 (2020): 453-466.

- Ringbauer, Harald, John Novembre, and Matthias Steinrücken. "Parental relatedness through time revealed by runs of homozygosity in ancient DNA." Nature communications 12.1 (2021): 1-11.

Reviewer #3: A review of Morez et al, Early medieval Pictish genomes reveal fine-scale relatedness in the UK.

The authors present a detailed examination of two Pictish genomes, using an array of tools and methods from the literature. This is a well written article that undertakes an informative and competent analysis of their data, and deserves to be published somewhere. They discuss interesting topics such as the potential for these societies to be matrilineal, and exactly who these people were.

However, the main sources of evidence are the amount of IBD that these two genomes share with other genomes, as well as a detailed discussion of seven mtDNA genomes. This is an interesting but slightly under-developed analysis - we could do the same for any data, and typically do not report the results in such detail, because it is fundamentally just a property of the data, and not enough to overcome the lack of sample diversity. The other methodology deployed is competent and appropriate but lacks detailed insight. The overall result then is somewhat anecdotal at the level of its implications for the diversity of historical Pictish cultural groups. Whilst the conclusions drawn are careful, and the topics discussed important, I therefore don't find this paper to be compelling enough for the highest tier of journal.

Technically I think this is a good paper without any substantial issues. The main suggestion, if the authors think it valuable and indeed possible, would be to try to find tools that are specifically designed to formalise the IBD and mt patterns into a model. Can we do any formal statistical testing for the mt results based on no-sex-bias assumptions? Counting IBD is something, but it is not e.g. LD-based admixture dating, which would really have helped say something new about the past. What does the distribution of IBD lengths tell us about the dates for which populations were in contact? etc. But maybe it is better to simply wait for more data to address these questions.

**Have all data underlying the figures and results presented in the manuscript been provided?**

Reviewer #1: Yes

Reviewer #2: **No: **Currently no ancient DNA data has been made available as of now. The raw data will be made available, but no processed ancient DNA data (underlying the figures) will be made available it seems.

Reviewer #3: Yes

PLOS authors have the option to publish the peer review history of their article (what does this mean?). If published, this will include your full peer review and any attached files.

Reviewer #1: **Yes: **Thomas Booth

Reviewer #2: **Yes: **Harald Ringbauer

Reviewer #3: No

---

## [Decision Letter · Decision Letter 1]

7 Feb 2023

Dear Dr Morez,

We are pleased to inform you that your manuscript entitled "Imputed genomes and haplotype-based analyses of the Picts of early medieval Scotland reveal fine-scale relatedness between Iron Age, early medieval and the modern people of the UK." has been editorially accepted for publication in PLOS Genetics. Congratulations!

While reviewers are enthusiastic about the revised manuscript, Reviewer 3 has made several suggestions, which we encourage you to consider in the final submission.  Before your submission can be formally accepted and sent to production you will need to complete our formatting changes, which you will receive in a follow up email. Please be aware that it may take several days for you to receive this email; during this time no action is required by you. Please note: the accept date on your published article will reflect the date of this provisional acceptance, but your manuscript will not be scheduled for publication until the required changes have been made.

Yours sincerely,

Jonathan Marchini

Academic Editor

PLOS Genetics

Hua Tang

Section Editor

PLOS Genetics

Comments from the reviewers (if applicable):

Reviewer's Responses to Questions

**Comments to the Authors:**

Reviewer #1: The authors have addressed my concerns, they. I'm very sorry to hear that the authors weren't able to access relevant data from Ireland, as the lack of such a comparison is a limitation to their study, but this is not their fault and the study is still publishable without it.

Reviewer #2: The authors did a thorough job addressing each of my comments. They clarified and expanded their article, and brought up valid reasons for their choices where they did not modify their approach. I believe that this article is now ready for publication with only minor edits. I congratulate the authors for their hard and good work, something they should be very proud of.

I have some last comments on their reply (A) and some minor suggestions regarding terminology (B). I stress that nothing of those are critical roadblocks.

Harald Ringbauer

A) Last comments

A1)

Thank you for running the additional qpAdm tests on the contaminated individuals.

The response and the new PCA picture (Fig. S10) clarified that even relatively small contamination with Native American ancestry such as the one of LUN003 (estimated 2-4%) can critically bias analysis when studying genetic fine-scale structure in Europeans. This reply clarifies that it was necessary to be conservative and to remove both individuals from downstream analysis, even though well below the usual contamination cutoffs of aDNA (which would apply for contamination with European ancestry and when looking for broad-scale patterns).

One interesting last thought is that it should be possible to "correct" the PCA projections in cases when the contaminating samples are known. PCA projection is "linear", i.e. a mixture of samples induces an additive mix of PC coordinates. Thus it is possible to subtract the right fraction of PC coordinates of the projected contaminant X to obtain the "uncontaminated" PC coordinates. The added uncertainty of fraction of contamination likely prohibits one to look into fine-scale genetic structure, but, if the authors feel it is worth their time, it could be interesting enough to do for their Westeurasian PCA in Fig. S10 (which is a very nice one!).

A2)

Thank you for adding a detailed reflection on potential bias in IBD calling (L336 - L359), you interacted very closely with the relevant literature and issues.

To not break the flow of the main text, which would distract a reader interested in the actual findings, I would recommend putting most of this "caveats" paragraph into

Identity-By-Descent and Homozygosity-By-Descent "Method" section (L545), only referring briefly to these limitations in the main text and pointing the interested reader to the method section.

B) Suggestions regarding Terminology

Given that the article is now close to publication as is, I have two minor recommendations regarding terminology. They are purely semantical. If the authors prefer to maintain the original nomenclature it would also be adequate.

B1) Throughout the manuscript, ages are often given in the "AD" notation (from Anno Domini):

To quote Wikipedia here: "A Terminology that is viewed by some as being more neutral and inclusive of non-Christian people is to call this the Current or Common Era (abbreviated as CE)"

However, it is certainly not unusual to use the "AD" notation in present-day publications.

B2) L262: Close-kin unions are referred to as "inbreeding"

Inbreeding often has very negative connotations. While being widely used as a technical term in plant and animal genomics, for humans the term "consanguinity" is often preferred when describing cousin unions or similar unions of rather close biological relatives.

Reviewer #3: The authors have done a strong job of addressing the concerns raised during the first round of review, and at a technical level I think this paper is well-undertaken and well-written. I remain somewhat on-the-fence about the significance of these results - primarily due to the small sample size, which mean that little at the population level can be said. It is ncongruous to on the one hand argue that Pictish people are not homogeneous, and on the other to generalise from one small geographical and temporal sample to all Pictish people.

That aside, I'm otherwise enthusiastic about the work, which is a very thorough investigation of the data available.

Minor point, there is still a little over-reach in the claims, with not enough caveats. Specific places:

1. Authors Summary "challenging older ideas that the Picts were a matrilineal society" needs a caveat as this is prominent. I don't think you can stretch this to "between groups during the Pictish period," but "this Pictish group".

2. p10 "We therefore suggest that individuals from Pictland should not be considered a homogenous genetic group, but instead a complex mixture of contemporary genetic ancestries." - I don't disagree that this is the most likely explanation, but we really don't have the sample size to separate migration, small population size, and wide-ranging population structure.

3. p18 "This finding challenges the older hypothesis that Pictish succession was based on a matrilineal system, assuming that wider Pictish society was organised in such a manner." also isolated from any caveats. I'd suggest "At least in this population group" or similar.

Very minor points for consideration:

Is Pictland a good word to use? Given that you argue (reasonably) that it isn't a unified thing. "Pictish people" or similar.

I don't think scientific naming should be held back by politics, but "British and Irish Isles" is a rare choice. You aren't consistent with it, with British being used alone in places.

**Have all data underlying the figures and results presented in the manuscript been provided?**

Reviewer #1: Yes

Reviewer #2: Yes

Reviewer #3: Yes

PLOS authors have the option to publish the peer review history of their article (what does this mean?). If published, this will include your full peer review and any attached files.

Reviewer #1: **Yes: **Thomas Booth

Reviewer #2: **Yes: **Harald Ringbauer

Reviewer #3: No

**Data Deposition**

http://datadryad.org/submit?journalID=pgenetics&manu=PGENETICS-D-22-00892R1

**Press Queries**

---

## [Editor Report · Acceptance letter]

31 Mar 2023

PGENETICS-D-22-00892R1 

Imputed genomes and haplotype-based analyses of the Picts of early medieval Scotland reveal fine-scale relatedness between Iron Age, early medieval and the modern people of the UK. 

Dear Dr Morez, 

We are pleased to inform you that your manuscript entitled "Imputed genomes and haplotype-based analyses of the Picts of early medieval Scotland reveal fine-scale relatedness between Iron Age, early medieval and the modern people of the UK." has been formally accepted for publication in PLOS Genetics! Your manuscript is now with our production department and you will be notified of the publication date in due course.

With kind regards,

Anita Estes

PLOS Genetics

On behalf of:
